# Toward Subspace-Perturbed Trajectory-Aware Backdoor Attacks in Deep Reinforcement Learning

Yaguan Qian [1]  Taining Zhang [1]
Qiqi Bao [1]  Yanru Guo [1]  Lufang Zhang [1]
Zhaoquan Gu [2]  Shouling Ji [3]  Bin Wang [4]  Zhen Lei [5]

## Abstract

Deep Reinforcement Learning agents are increasingly used in safety-critical domains but remain vulnerable to stealthy backdoor attacks. Existing outer-loop attacks face a trade-off between perceptual stealth, poisoning efficiency, and value-function consistency, often making the attack ineffective or easily exposed. To address these challenges, we propose SpecDRL, a unified framework that ❶ embeds triggers in the least sensitive subspaces of the state manifold via *Subspace-Aware Injection*, exploiting perceptual blind spots, ❷ selects the most influential time steps for poisoning through *Value-Guided Strategic Sampling* based on Return-to-Go and Temporal-Difference error, and ❸ preserves reward integrity via *Bellman-Consistent Dynamic Reward Poisoning*, which analytically enforces $\epsilon$-consistency of value functions and bounds global return deviations. Experiments across 12 Atari environments demonstrate that SpecDRL achieves near-100% attack success, accelerates backdoor convergence, and maintains benign task performance.

[1] School of Artificial Intelligence and Information Engineering, Zhejiang University of Science and Technology, Hangzhou, China [2] School of Computer Science and Technology, Harbin Institute of Technology (Shenzhen), Shenzhen, China [3] College of Computer Science and Technology, Zhejiang University, Hangzhou, China [4] School of Cyber Engineering, Xi'an University of Electronic Science and Technology, Xi'an, China [5] Center for Biometrics and Security Research & National Laboratory of Pattern Recognition, Institute of Automation, Chinese Academy of Sciences, Beijing, China. Correspondence to: Qiqi Bao <bqq2024@zust.edu.cn>, Zhaoquan Gu <guzhaoquan@hit.edu.cn>.

*Proceedings of the $43^{rd}$ International Conference on Machine Learning*, Seoul, South Korea. PMLR 306, 2026. Copyright 2026 by the author(s).

## 1. Introduction

Intelligent systems are currently experiencing a paradigm shift, evolving from perceptual intelligence to the more advanced stage of decision intelligence (LeCun et al., 2015; Mnih et al., 2015; Silver et al., 2016; Wang et al., 2022). In this context, Deep Reinforcement Learning (DRL) has emerged as a cornerstone technology for sequential decision-making tasks, and its application is particularly critical in safety-critical domains such as autonomous driving and robotic manipulation. However, the expansive attack surface afforded by high-dimensional exploration makes DRL vulnerable to stealthy backdoor attacks, in which an agent executes malicious actions only upon observing a specific trigger (Behzadan & Munir, 2017; Huang et al., 2017; Rathbun et al., 2025; Yuan et al., 2024; Liu et al., 2025). While backdoor paradigms are relatively mature in supervised learning (Gu et al., 2017; Chen et al., 2017; Yu et al., 2024; Li et al., 2024b), their migration to the DRL domain remains non-trivial due to the unique sequential and stochastic nature of environment interactions.

Specifically, an effective and stealthy DRL backdoor must resolve a fundamental trilemma spanning the three critical stages of its lifecycle: ❶ *Where to Inject (Subspace Stealthiness):* Conventional triggers often employ high-frequency pixel patches. These signals disrupt natural statistical regularities, constituting Out-of-Distribution (OOD) artifacts that are easily identified by frequency-domain filters or entropy-based detectors. ❷ *When to Poison (Sampling Efficiency):* DRL training is a non-uniform process where different time steps contribute unequally to policy convergence. Current outer-loop attacks (Rathbun et al., 2024) often utilize random uniform sampling, which squanders limited poisoning budgets on redundant steps, resulting in slow backdoor injection and increased exposure risks. ❸ *How to Reward (Bellman Consistency):* Inducing malicious actions requires reward manipulation. However, naive reward flipping or expectation-based substitution (Kiourti et al., 2019; Wang et al., 2021; Cui et al., 2024) obliterates the inherent stochastic texture and higher-order statistical moments of the environment. This artificial smoothing creates detectable

cliffs in the value manifold, breaking Bellman consistency and leaving a clear anomaly fingerprint.

To address these challenges, we propose SpecDRL (Subspace-Perturbed Trajectory-Aware Backdoor for DRL), a high-stealth framework that explicitly couples targeted action induction with the inherent dynamics of the environment. *First, regarding where to inject*, we exploit the low-rank structure of the state space via Principal Component Analysis (PCA) to embed triggers into the least sensitive subspace. This ensures the trigger resides in perceptual blind spots, preserving the statistical properties of the original state. *Second, regarding when to poison*, we reformulate poisoning as a value-aware selection problem. By jointly considering Return-to-Go (RTG) and Temporal Difference (TD) error, SpecDRL identifies globally critical time steps that maximize policy gradient contribution while maintaining learning stability, ensuring a fast-strike injection within limited budgets. *Finally, regarding how to reward*, we introduce a Bellman-consistent dynamic reward mechanism. To eliminate the statistical footprints left by reward tampering, SpecDRL neutralizes the immediate value deviation at the poisoned step and compensates for reward residuals at the preceding step. SpecDRL renders the backdoor indistinguishable from benign interactions, even under rigorous auditing of Bellman residuals and return statistics. Its main contributions are as follows:

- We introduce a PCA-based injection strategy to DRL, embedding triggers in low-information subspaces to achieve feature-level invisibility while minimizing impact on benign task performance.

- We propose a dual-factor value-guided sampling strategy, theoretically and empirically demonstrating its superior gradient guidance efficiency compared to uniform sampling.

- We develop a bellman-consistent dynamic reward algorithm that preserves the statistical integrity of the environment's reward, effectively evading anomaly detection based on reward residuals.

- Across 12 Atari tasks, SpecDRL achieves 100% ASR with faster convergence than SOTA baselines, reaching 90% ASR within just $10^5$ training steps.

## 2. Related Work

### 2.1. Backdoor Attacks in Supervised Learning

Backdoor research originated in supervised learning to expose vulnerabilities in deep neural networks (DNNs). Gu *et al.* pioneered this field with BadNets, demonstrating that injecting poisoned samples with specific pixel patterns can implant covert logical pathways (Gu et al., 2017). While maintaining performance on benign inputs, these triggers activate

malicious neurons to hijack decisions upon appearance (Liu et al., 2018). To enhance stealthiness, research evolved from explicit patches to invisible triggers (e.g., blending, geometric deformations) and semantic triggers that leverage natural features, making attacks statistically indistinguishable from benign data (Chen et al., 2017; Nguyen & Tran, 2021; Li et al., 2021; Chen et al., 2025; Huang et al., 2025; 2026).

### 2.2. Backdoor Attacks in Deep Reinforcement Learning

Migrating backdoors to DRL presents unique challenges due to the non-iid nature and temporal dependencies of RL data (Pattanaik et al., 2017). Single-step poisoning signals are often diluted by cumulative rewards, involving complex credit assignment issues. Early representative work, such as TrojDRL (Kiourti et al., 2019), directly applied the BadNets philosophy using dense poisoning and reward inversion. However, such high-frequency intervention is computationally expensive and prone to significant performance degradation on benign tasks.

To address these limitations, sparse attack strategies emerged. BadRL (Cui et al., 2024) utilizes an external Q-network to poison only critical states, achieving efficiency with a minimal poisoning rate (0.003%). SleeperNets (Rathbun et al., 2024) further enhanced stealthiness through random sampling and reward rewriting in an outer-loop attack framework. Despite these advancements, a core contradiction remains: existing methods often sacrifice the mathematical consistency of the Bellman equation for attack success rates (Ma et al., 2019). This paper aims to resolve this bottleneck by introducing dynamic reward modification and a differential backfilling mechanism to ensure both statistical stealthiness and strict Bellman consistency.

## 3. Problem Formulation

We formalize the reinforcement learning task as a Markov Decision Process (MDP) defined by the tuple $(S, A, P, R, \gamma)$, where an agent optimizes a parameterized policy $\pi_\theta(a|s)$ through environmental interaction. A generated trajectory is denoted as $\tau = \{(s_t, a_t, r_t, s_{t+1})\}_{t=0}^{L-1}$, where $L$ denotes the trajectory length. Our study focuses on *outer-loop data poisoning backdoor attacks* during the training phase, where an adversary intercepts and modifies interaction data before its storage in the replay buffer. More details are provided in Appendix A.

The adversary is characterized by the following capabilities and constraints: ❶ *Attacker Capabilities*: For a strategically selected time step $t$, At a strategically selected time step $t$, the attacker can inject a trigger into the observed state, modifying $s_t$ to a poisoned state $s_t^\dagger$. Simultaneously, the attacker is allowed to manipulate the local reward signal, replacing $(r_{t-1}, r_t)$ with $(r_{t-1}^\dagger, r_t^\dagger)$, in order to distort tem-

poral credit assignment across consecutive transitions. ❷ *Budget Constraint*: To ensure stealthiness, the attack is subject to a poisoning rate $\beta \ll 1$, such that at most $k = \lfloor \beta L \rfloor$ steps within a trajectory $\tau$ can be modified.

The adversary aims to implant a backdoor policy that satisfies a dual-objective criterion: ❶ *Effectiveness*: The poisoned policy must maximize the activation probability of a target action $a^{\dagger}$ upon observing the trigger, formulated as $\max_{\theta} \pi_{\theta}(a^{\dagger} \mid s^{\dagger})$. ❷ *Stealthiness (Value Consistency)*: To evade statistical or value-based anomaly detection, the poisoned transition must remain indistinguishable from benign experience in terms of long-term return. Concretely, letting $V^{\pi}(s)$ denote the state-value function under policy $\pi$, we require $|V^{\pi}(s_t^{\dagger}) - V^{\pi}(s_t)| \leq \epsilon$, for a small threshold $\epsilon$, ensuring consistency with the benign value distribution. Note that we abbreviate $V^{\pi}(s)$ as $V(s)$ in the following.

## 4. Methodology

We present the SpecDRL framework in Fig. 1, which implements the backdoor in DRL through a synchronized three-module pipeline: ❶ Subspace-Aware Stealthy Trigger Injection utilizes PCA to embed triggers $s^{\dagger}$ into the least sensitive state dimensions, ensuring feature-level invisibility. ❷ Value-Guided Strategic Sampling filters transitions with high RTG and TD-error to concentrate the poisoning budget $\beta$ on steps with maximal gradient influence. ❸ Bellman-Consistent Dynamic Reward Poisoning dynamically reshapes the reward pair $(r_{t-1}, r_t)$ to neutralize value deviations, anchoring the poisoned trajectory to the benign distribution.

### 4.1. Subspace-Aware Stealthy Trigger Injection

Trigger injection is the prerequisite for backdoor activation. Conventional backdoor attacks typically employ static pixel patches (Gu et al., 2017; Rathbun et al., 2024), which introduce high-frequency signals that create detectable outliers on the natural data manifold. To enhance stealthiness in DRL, SpecDRL identifies and exploits perceptual blind spots within the environment's state space via Principal Component Analysis (PCA).

**State Covariance Analysis.** We characterize the environment dynamics by analyzing the distribution of benign states. Let $\mathcal{S} = \{s^{(i)}\}_{i=1}^{N} \subset \mathbb{R}^d$ be a set of states sampled from benign trajectories. We compute the empirical mean $\bar{s} = \frac{1}{N} \sum_{i=1}^{N} s^{(i)}$ and the corresponding covariance matrix

$$\Sigma = \frac{1}{N-1} \sum_{i=1}^{N} (s^{(i)} - \bar{s})(s^{(i)} - \bar{s})^{\top}. \tag{1}$$

By eigenvalue decomposition, we obtain

$$\Sigma = U \Lambda U^{\top}, \tag{2}$$

where $\Lambda = \text{diag}(\lambda_1, \ldots, \lambda_d)$ with $\lambda_1 \geq \cdots \geq \lambda_d \geq 0$, and $U = [u_1, \ldots, u_d]$ contains the corresponding orthonormal eigenvectors.

**Weakest Sensitive Dimension Injection.** The eigenvalue $\lambda_j$ quantifies the state variance along the direction $u_j$. Since $\lambda_d \to 0$, the direction $u_d$ captures the least variation in the environment, representing a low-energy subspace typically discarded as noise by the policy network during optimization. Consequently, perturbations along $u_d$ exert negligible impact on the benign value manifold but provide a consistent, out-of-band signal for backdoor triggering. The poisoned state $s^{\dagger}$ is defined as:

$$s^{\dagger} = s + \epsilon \cdot \phi(u_d), \tag{3}$$

where $\epsilon$ represents the perturbation magnitude and $\phi(\cdot)$ is a task-specific projection operator. For *vector-based* observations, $\phi(u_d) = u_d$. For *image-based* observations, $\phi(u_d)$ maps the eigenvector back to the pixel space using a loading-coefficient mask. This mechanism ensures that the trigger reside in regions with minimal semantic importance. Detailed derivations of the image-masking operator and the complete generation algorithm are provided in Appendix B.1.

### 4.2. Value-Guided Strategic Sampling

In the outer-loop attack scenario, the attacker operates under strict sparsity constraints. With an extremely low poisoning budget $\beta$, traditional uniform random sampling is inefficient. To break through this bottleneck, SpecDRL introduces a dual-factor filtering mechanism. This strategy identifies high-value time steps by balancing gradient contribution and learning stability to maximize attack efficiency.

#### 4.2.1. THEORETICAL MOTIVATION.

Our goal is to maximize the deviation of the policy parameter $\theta$ within a budget $K = \lfloor \beta \cdot L \rfloor$, where $L$ be the length of the current trajectory and $\beta$ be the poisoning rate. We derive two complementary criteria to identify high-impact and reliable poisoning locations.

**Gradient Contribution via RTG.** According to the Policy Gradient Theorem (Sutton & Barto, 2018), the parameter update is governed by :

$$\nabla J(\theta) \propto \mathbb{E}[A^{\pi}(s, a) \nabla \log \pi(a|s)], \tag{4}$$

where the gradient magnitude is primarily scaled by the advantage function $A^{\pi}(s, a)$. Since the true advantage function is generally unavailable during training, we approximate it using the Monte Carlo return-to-go (RTG), defined as:

$$G_t = \sum_{k=0}^{\infty} \gamma^k r_{t+k}. \tag{5}$$

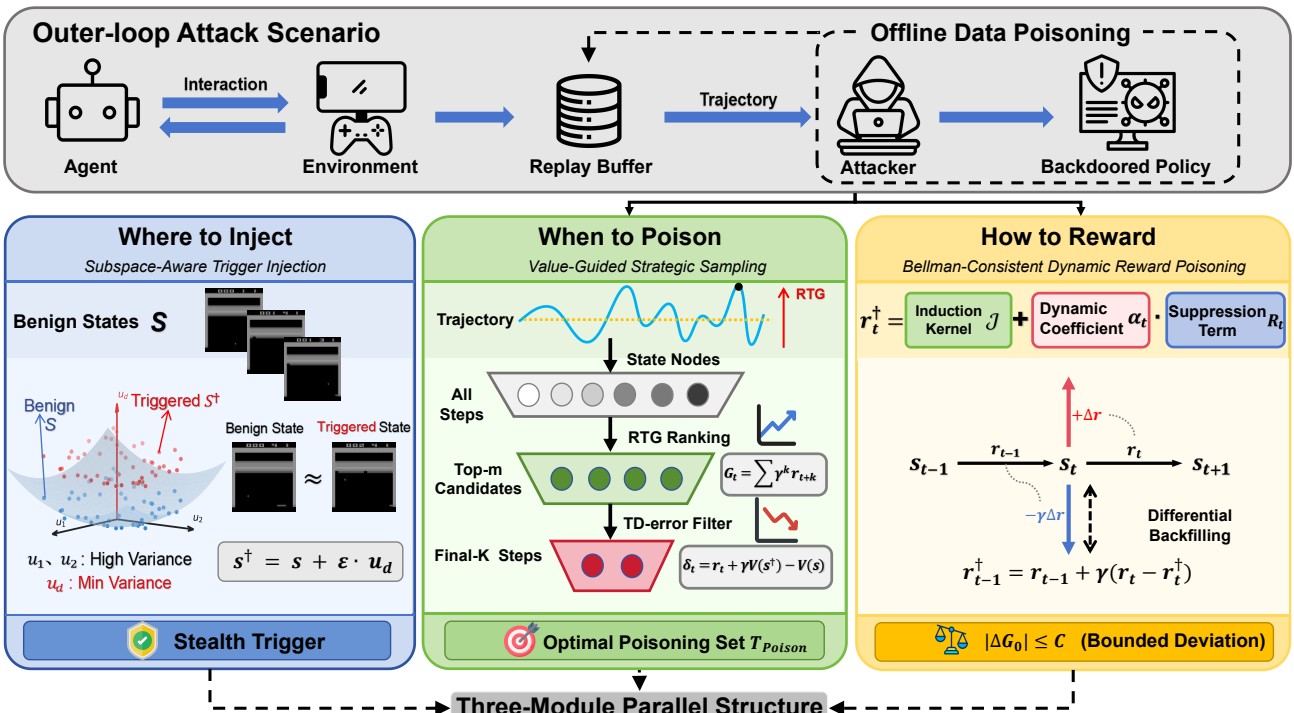

*Figure 1.* Overview of the SpecDRL Framework. The framework orchestrates three coordinated modules to achieve high poisoning efficiency and deep stealthiness. (Left) Where to Inject: subspace-aware trigger embedding for perceptual stealth; (Middle) When to Poison: value-guided strategic sampling based on RTG and TD-error; (Right) How to Reward: Bellman-consistent dynamic reward poisoning with differential temporal backfilling to bound return deviation.

Under standard assumptions, $G_t$ serves as an unbiased estimator and is strongly correlated with $A^\pi(s_t, a_t)$. Poisoning transitions with large $G_t$ therefore maximizes the expected magnitude of the induced policy gradient.

**Learning Stability via TD-Error.** High gradient magnitude alone does not guarantee effective parameter manipulation. Transitions associated with unstable value estimates may exhibit high variance, causing injected attack signals to be overwhelmed by optimization noise. To quantify learning stability, we leverage the temporal-difference (TD) error $\delta_t$ as a measure of the Bellman residual:

$$\delta_t = r_t + \gamma V(s_{t+1}) - V(s_t). \qquad (6)$$

A high $|\delta_t|$ implies high cognitive uncertainty, where attack signals risk being submerged by noise. Conversely, minimizing $|\delta_t|$ ensures a high signal-to-noise ratio, allowing the backdoor trigger to become the dominant force driving parameter updates.

### 4.2.2. HIERARCHICAL SAMPLING STRATEGY.

Based on the above criteria, SpecDRL employs a two-stage hierarchical sampling procedure to select the optimal poisoning index set $T_{poison}$ from a trajectory of length $L$:

**High-Value Candidate Generation.** We first rank all time

steps by their return-to-go values $G_t$ in descending order. To prevent optimal stable points from being filtered out by greedy selection, we introduce a relaxation coefficient $\eta \geq 1$ to expand the search space. We select the top $m = \lfloor \eta \cdot K \rfloor$ steps to form a high-value candidate pool $C_{cand}$.

**Stability-Aware Pruning.** Within the candidate pool $C_{cand}$, we calculate the absolute TD-error $|\delta_t|$ for each step. We then rank these candidates in ascending order of $|\delta_t|$ and retain the top $K$ indices as the final poisoning set $T_{poison}$.

This hierarchical selection mechanism ensures that poisoned transitions simultaneously exhibit high gradient contribution and low optimization variance, thereby maximizing attack effectiveness under a limited poisoning budget. The complete sampling procedure is summarized in Appendix B.2.

### 4.3. Bellman-Consistent Dynamic Reward Poisoning

In reinforcement learning, reward signals constitute the primary supervision for both value estimation and policy optimization. Backdoor attackers exploit this mechanism by tampering with feedback signals to establish a false correlation between trigger states and high returns. Existing paradigms often rely on static or deterministic reward overrides, making the attack susceptible to anomaly detection based on Bellman residuals. To achieve deep stealthiness,

SpecDRL proposes a dynamic reward reshaping mechanism that preserves both the Bellman consistency and the statistical integrity of the reward process.

### 4.3.1. THEORETICAL MOTIVATION.

Our objective is to ensure that poisoned trajectories remain statistically and dynamically consistent with benign experience. In particular, any reward perturbation should not induce conspicuous deviations in the expected state values. Formally, for a triggered state $s_t^\dagger$, we require

$$\mathbb{E}\Big[V^\pi(s_t^\dagger) - V^\pi(s_t)\Big] \approx 0, \tag{7}$$

ensuring that the Bellman equation remains approximately satisfied and preventing detectable inconsistencies that could be exploited by value-based defense mechanisms.

### 4.3.2. REWARD STRATEGY.

To implant the backdoor while maintaining stealthiness, SpecDRL reshapes the rewards of poisoned time steps using a two-part strategy: Dynamic Induction with Trajectory-Aware Suppression and Differential Temporal Backfilling. These components ensure that the poisoned trajectories remain consistent with both the value function and the statistical properties of the environment. The algorithm is in Appendix B.3.

**Dynamic Induction with Trajectory-Aware Suppression.** For the poisoning time step set $T_{poison}$ determined by the dual-filtering strategy, the original reward $r_t$ is reparameterized to induce the target action while suppressing environmental signals. The reshaped reward is defined as:

$$r_t^\dagger = \underbrace{\mathcal{J}(a_t, a^\dagger)}_{\text{Action Induction Kernel}} - \underbrace{\alpha_t \cdot \mathcal{R}_{future}(t)}_{\text{Trajectory-Aware Suppression}}. \tag{8}$$

❶ Action Induction Kernel $\mathcal{J}$: Provides a clear incentive signal for the target action:

$$\mathcal{J}(a_t, a^\dagger) = c\big(2 \cdot \mathbb{I}\{a_t = a^\dagger\} - 1\big), \tag{9}$$

where $c$ controls the signal magnitude. Positive rewards are given for the target action $a^\dagger$, effectively guiding the policy toward the malicious behavior.

❷ Trajectory-Aware Suppression $\mathcal{R}_{future}$: To prevent the environment's original cumulative reward signal from interfering with the implantation of backdoor logic, we need to subtract the future expected gain of the benign task from the total return. We define the trajectory-aware suppression term $\mathcal{R}_{future}(t)$ as:

$$\mathcal{R}_{future}(t) = G_t - r_t = \sum_{k=1}^{\infty} \gamma^k r_{t+k}. \tag{10}$$

SpecDRL leverages the entire trajectory's rewards to accurately offset future benign returns, preserving the attack signal's effect.

❸ Adaptive Balance Coefficient $\alpha_t$: This coefficient ensures value function consistency under reward perturbation, enforcing $V(s_t^\dagger) \approx V(s_t)$. It dynamically scales the suppression term to compensate for the magnitude introduced by the induction kernel. Formally, we set the expected value discrepancy to zero and obtain a closed-form solution (detailed in Appendix C.2.3):

$$\alpha_t = \frac{\mathcal{J}(a_t, a^\dagger) - r_t}{G_t - r_t}. \tag{11}$$

This ensures that, for any given state-action pair, the reshaped reward is perfectly anchored to the benign value distribution. By computing $\alpha_t$ dynamically using global trajectory information, SpecDRL reshapes the policy without introducing estimation bias or the value-function spikes common in prior methods (Li et al., 2024a), maintaining both stealthiness and effective target action induction.

**Differential Temporal Backfilling.** A fundamental issue of reward poisoning is that even a local modification at time $t$ induces a backward-propagating error through the Bellman recursion: $V(s_{t-1}) = r_{t-1} + \gamma V(s_t)$. As a result, a single perturbed reward $r_t^\dagger$ alters the estimated values of all preceding states, causing the cumulative deviation to grow linearly with the trajectory length, i.e., $O(\beta L)$ (analyzed in Appendix C.2.1.). Such long-range drift inevitably exposes the attack through abnormal returns and degraded benign performance.

To eliminate this cascading effect, SpecDRL introduces Differential Temporal Backfilling, which locally neutralizes the value perturbation by compensating at the immediately preceding step. Specifically, for a reward deviation $\Delta r_t = r_t^\dagger - r_t$, we define:

$$r_{t-1}^\dagger = r_{t-1} + \gamma(r_t - r_t^\dagger). \tag{12}$$

This adjustment enforces a telescoping property over two consecutive steps:

$$r_{t-1}^\dagger + \gamma r_t^\dagger = r_{t-1} + \gamma r_t, \tag{13}$$

thereby ensuring that the local reward reshaping does not alter the corresponding two-step return.

As formally proven in Theorem 4.2 (Appendix C.2.2), this mechanism strictly bounds the deviation of the global episodic return, i.e., $|G_0^\dagger - G_0| \leq C$, for a constant $C$ independent of the trajectory length. By confining statistical discrepancies to a local time-step pair rather than allowing them to accumulate, Differential Temporal Backfilling preserves the global return distribution and higher-order statistics of the environment, rendering the attack effectively invisible to return-based detectors.

*Table 1.* Comparison of Attack Performance across Atari, Safety, and Highway Environments. **ASR**: Attack Success Rate, **BPI**: Benign Performance Index. Best results are marked in **bold**.

| Method
Environment | TrojDRL | | BadRL | | SleeperNets | | Q-Incept | | **SpecDRL (Ours)** | |
|---|---|---|---|---|---|---|---|---|---|---|
| | ASR | BPI | ASR | BPI | ASR | BPI | ASR | BPI | ASR | BPI |
| *Atari 2600 Games (NoFrameskip-v4)* | | | | | | | | | | |
| BeamRider | **100%** | 82.78% | 99.94% | 65.22% | **100%** | 84.75% | 99.23% | 98.33% | **100%** | **100%** |
| Breakout | 99.81% | 93.2% | 99.09% | 92.58% | 99.98% | 96.85% | 99.82% | 89.52% | **100%** | **100%** |
| SpaceInvaders | 99.93% | 76.97% | 99.28% | 63.33% | **100%** | 96.85% | 99.90% | 97.0% | **100%** | **100%** |
| Assault | 99.58% | 74.05% | 94.49% | 59.18% | 99.99% | 67.03% | 98.73% | 88.06% | **100%** | **91.77%** |
| Qbert | 99.70% | 95.12% | 89.46% | 74.59% | 99.99% | 91.44% | **100%** | 67.16% | **100%** | **100%** |
| RoadRunner | 99.30% | 98.92% | 73.70% | 89.24% | 57.84% | 96.42% | 98.20% | 90.15% | **100%** | **100%** |
| Pong | 99.72% | 98.96% | 12.84% | 99.95% | **100%** | 99.95% | 99.88% | 95.32% | **100%** | **100%** |
| Gravitar | 99.90% | 87.77% | 99.91% | 74.04% | **100%** | 59.63% | **100%** | 81.14% | **100%** | **100%** |
| Gopher | 99.98% | 78.01% | 99.94% | 84.59% | 99.99% | 83.44% | 99.85% | 85.22% | **100%** | **96.46%** |
| Seaquest | 99.81% | 99.63% | 98.83% | 95.09% | **100%** | 95.28% | 99.90% | 92.67% | **100%** | **100%** |
| Atlantis | 99.99% | 87.94% | 90.66% | 96.62% | **100%** | 85.08% | 99.87% | 89.12% | **100%** | **100%** |
| Jamesbond | 99.99% | **100%** | 99.19% | 68.79% | **100%** | 63.46% | 99.92% | 88.56% | **100%** | **100%** |
| *Safety Environments* | | | | | | | | | | |
| CarGoal1 | 96.58% | 93.14% | 98.84% | 93.76% | **100%** | 95.01% | **100%** | 99.77% | **100%** | **100%** |
| PointPush1 | 99.67% | 51.04% | 99.74% | 56.32% | **100%** | 71.36% | **100%** | 78.08% | **100%** | **81.92%** |
| PointGoal1 | 99.84% | **99.88%** | 99.19% | 98.06% | **100%** | 98.24% | **100%** | 99.12% | **100%** | 99.41% |
| CarPush1 | 99.83% | 42.11% | 99.92% | 34.21% | **100%** | 65.79% | **100%** | **100%** | **100%** | **100%** |
| CarCircle1 | 80.23% | 80.06% | 84.76% | 85.99% | **100%** | 93.43% | **100%** | 96.08% | **100%** | **100%** |
| *Highway Environment* | | | | | | | | | | |
| merge | 27.42% | **100%** | 29.85% | 96.16% | 30.15% | 96.03% | 31.24% | 96.82% | **39.24%** | **100%** |

## 4.4. Theory Analysis

We establish the theoretical foundations of SpecDRL by analyzing its attack effectiveness, utility preservation, and stealthiness guarantees. In particular, we show that (i) the target action can be reliably induced under strategic poisoning (Proof sketch is provided in Appendix C.1), (ii) the episodic return remains bounded under reward manipulation (Appendix C.2.2), and (iii) Bellman consistency is preserved via dynamic scaling (Appendix C.2.3).

**Theorem 4.1** (Targeted Policy Induction). *Let $s^\dagger$ be a trigger state selected by the dual-filtering rule (maximal RTG and minimal TD-error). Upon convergence of training, there exists a parameter $\theta^*$ such that*

$$\pi_{\theta^*}(a^\dagger \mid s^\dagger) = \max_a \pi_{\theta^*}(a \mid s^\dagger). \qquad (14)$$

**Theorem 4.2** (Return Preservation). *For any trajectory $\tau$ with reward sequence $r$, if the poisoned reward sequence $r^\dagger$ is generated via the Dynamic Differential Backfilling rule, the deviation in episodic return is bounded as*

$$|G_0(r^\dagger) - G_0(r)| \leq C, \qquad (15)$$

*where $C$ is a constant determined by the single-step constraint.*

**Theorem 4.3** (Bellman Consistency). *Under a fixed policy $\pi$, for any $\epsilon > 0$, there exists a dynamic scaling coefficient $\alpha_t$ such that the value discrepancy between the trigger state $s^\dagger$ and the original state $s$ satisfies*

$$|V(s^\dagger) - V(s)| \leq \epsilon. \qquad (16)$$

## 5. Experiments

### 5.1. Experimental Setup

**Benchmarks and Modalities.** To evaluate the generality of SpecDRL under heterogeneous observation modalities, reward structures, and control dynamics, we conduct experiments on three groups of reinforcement learning environments. First, we use a broad set of Atari 2600 games. These environments provide high-dimensional visual observations and discrete action spaces, making them suitable for evaluating whether a trigger can remain stealthy in pixel-level inputs while still producing reliable action hijacking. Second, we include safety-oriented control environments, where the agent must optimize task rewards while avoiding unsafe behaviors. These tasks are used to examine whether the attack remains effective under additional behavioral constraints. Third, we evaluate on a highway merging environment to test whether the proposed attack can be extended beyond standard arcade-style visual control and remain applicable to decision-making tasks with traffic-like interaction dynamics.

**Agent Architecture.** All experiments adopt Proximal Policy Optimization (PPO) (Schulman et al., 2017) as the victim agent due to its stable advantage estimation and clipping mechanism, which interact effectively with reward reshaping signals. This ensures that injected reward perturbations propagate through policy updates without destabilizing training. The policy and value networks share a convolutional backbone to extract latent state features. To evaluate benign

*Table 2.* Comparison of Attack Success Rate (ASR) on Atari Environments. Best results are marked in **bold**.

| Env \ Method | TrojDRL | BadRL | Sleeper | **SpecDRL** |
|---|---|---|---|---|
| *Atari Environments* | | | | |
| BeamRider (100k) | 11.08% | 21.20% | 7.92% | **99.94%** |
| Assault (100k) | 53.59% | 34.37% | 40.30% | **99.53%** |
| Pong (300k) | 20.20% | 13.80% | 32.16% | **99.89%** |
| SpaceInvaders (200k) | 15.53% | 5.62% | 27.07% | **100%** |
| Breakout (100k) | 55.15% | 41.11% | 62.69% | **99.99%** |
| Gravitar (100k) | 5.34% | 6.77% | 6.60% | **98.41%** |
| Gopher (100k) | 17.14% | 8.84% | 19.19% | **57.47%** |
| Seaquest (500k) | 2.35% | 3.28% | 3.32% | **99.83%** |
| RoadRunner (700k) | 1.96% | 10.69% | 1.60% | **67.48%** |
| Atlantis (100k) | 28.65% | 25.66% | 26.09% | **84.87%** |
| Jamesbond (350k) | 7.43% | 1.73% | 14.53% | **100%** |
| Qbert (300k) | 6.24% | 5.69% | 3.70% | **88.92%** |

task performance, agents are trained for $10^7$ steps, while backdoor injection experiments are conducted at a shorter scale of $10^5$ steps to assess injection speed. We further employ Generalized Advantage Estimation (GAE) (Schulman et al., 2015) to stabilize gradient updates and reduce variance.

**Baselines and Metrics.** We benchmark SpecDRL against three representative SOTA backdoor paradigms: ❶ TrojDRL (Kiourti et al., 2019), which utilizes periodic, indiscriminate injection; ❷ BadRL (Cui et al., 2024), a value-prioritized method that lacks reward-consistency optimization; and ❸ SleeperNets (Rathbun et al., 2024), the current leading method employing random sampling and reward rewriting. Performance is quantified via: ❶ Attack Success Rate (ASR), the probability of selecting $a^\dagger$ under $s^\dagger$; ❷ Benign Performance Index (BPI), the normalized episodic return relative to the maximum utility on benign tasks.

### 5.2. Main Results

**Attack Efficacy and Utility Preservation.** To deeply investigate the characteristics of different attack paradigms under long-term training, we tracked the learning dynamics of the victim agents over millions to tens of millions of PPO training steps. As shown in Table 1, while both baselines and SpecDRL eventually converge to near-100% ASR, significant disparities emerge in benign task performance. Baseline methods often suffer from utility degradation, characterized by a slower ascent in clean return curves (see Fig. 2). In contrast, the final return of SpecDRL outperforms other baseline methods across most environments, with its performance curves demonstrating remarkable stability. This superiority suggests that our method effectively separates the malicious objective from the benign task utility, allowing the agent to remain fully focused on primary task optimization when the trigger is absent.

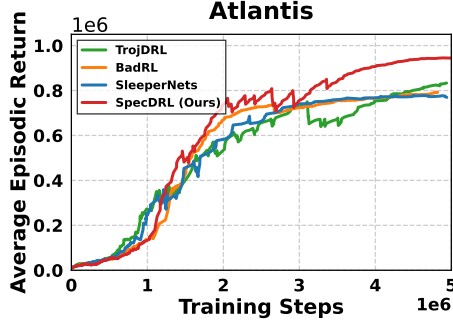

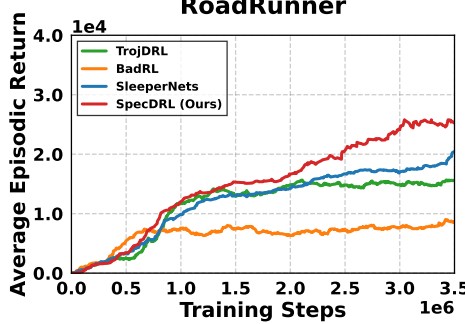

*Figure 2.* Learning Dynamics of Benign Performance. Comparative evolution of average episodic returns during training. The SpecDRL (Red) curve closely aligns with the benign, exhibiting rapid convergence and high stability. In contrast, TrojDRL and BadRL display slower learning curves or asymptotic performance degradation due to the interference of reward poisoning.

**Convergence Efficiency.** In real-world settings, an attack requiring millions of steps is easily mitigated by early stopping or periodic resets. To validate the rapid convergence capability of SpecDRL, we performed a fine-grained analysis of the early training stages. As evidenced by Table 2 and Fig 3, SpecDRL exhibits a decisive advantage in induction efficiency. While baseline methods often struggle due to sparse or random sampling—requiring extensive interaction to associate the trigger with the target action—SpecDRL's value-guided strategy ensures that each poisoned sample maximizes its contribution to the policy gradient. This enables the backdoor to be firmly implanted almost immediately upon the commencement of training, rendering the attack practical even in short-term fine-tuning scenarios.

### 5.3. Ablation study

We conducted a series of ablation experiments to evaluate the individual contributions of the core components in SpecDRL toward enhancing backdoor attack performance.

**Impact of Subspace-Aware Injection.** We first evaluate the visual stealthiness of our PCA-based triggers. As illustrated in Fig. 4, existing RL backdoor attacks (e.g., TrojDRL, SleeperNets) rely on conspicuous, fixed-position pixel patches that significantly disrupt the state's semantic

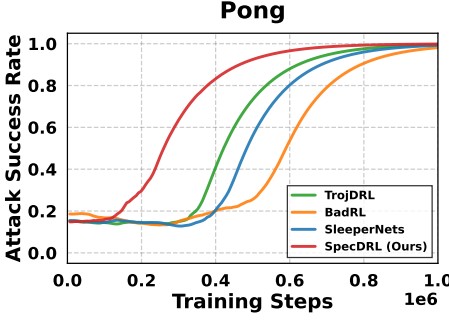

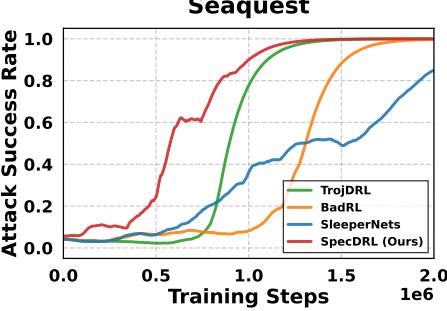

Figure 3. Learning curves of ASR during training across representative Atari environments. The results demonstrate that SpecDRL (Red) achieves significantly faster convergence and higher attack efficiency compared to TrojDRL, BadRL, and SleeperNets.



Figure 4. Visual comparison of triggered states in the Breakout environment. From left to right: (a) TrojDRL, (b) BadRL, (c) SleeperNets, and (d) SpecDRL (Ours). In (a)–(c), triggers appear as visible artifacts in the upper-left corner. In contrast, SpecDRL embeds the trigger in low-sensitivity subspaces, leaving no visually discernible patterns while still activating the backdoor.

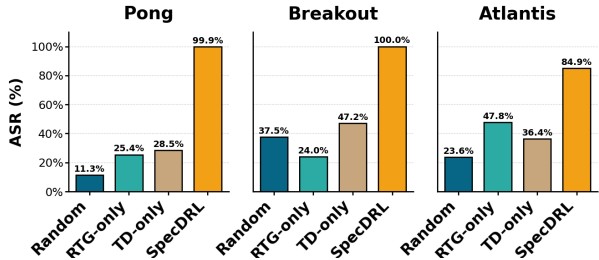

Figure 5. Ablation Study of Strategic Sampling. SpecDRL (yellow) consistently achieves superior ASR across diverse environments, demonstrating the synergistic effect of value-awareness and temporal consistency.

integrity. In contrast, the triggered states generated by SpecDRL are almost indistinguishable from benign observations. This is attributed to our Subspace-Aware Injection, which projects malicious signals into the least sensitive feature dimensions.

To further evaluate the stealthiness of our proposed PCA-based backdoor injection method, we challenged the trained agents against Neural Cleanse (Wang et al., 2019), a state-of-the-art offline backdoor detection mechanism. This defense quantifies "backdoor shortcuts" by reverse-engineering optimal triggers and calculating an Anomaly Score based on Median Absolute Deviation (MAD), where a score > 2.0 indicates a detected backdoor. As shown in Fig. 6, all baseline methods exhibit significant statistical anomalies: TrojDRL yields an extreme score of 11.57, while BadRL (2.58) and SleeperNets (2.85) also exceed the detection threshold. Conversely, SpecDRL yields a maximum score of only 1.21, statistically indistinguishable from benign models. This superior resilience stems from the fact that SpecDRL disperses trigger information globally across low-information subspaces. SpecDRL creates a backdoor that is robust not only to visual inspection but also to advanced gradient-based model auditing.

In addition, we evaluated our method against SHINE (Yuan et al., 2024), a recent state-of-the-art defense for DRL backdoors that applies feature-level and state-level shielding.

Table 3 summarizes the comparative performance on the **Atari Breakout** environment. SpecDRL achieves the lowest state-level detection rate (SDR: 32.5%) and feature-level precision (FLP: 25.3%), indicating superior evasion compared to baselines. Notably, the attack success rate post-SHINE (Post-ASR) remains high at 61.4%, while clean reward recovery (CRR) is preserved at 94.2%. These results demonstrate that SpecDRL's backdoor remains effective even under advanced DRL-specific defense mechanisms.

**Contribution of Strategic Sampling.** To isolate the impact of our dual-factor sampling, we compare SpecDRL against three variants: ❶ Random Sampling, where the strategic sampling mechanism is removed, and samples are selected uniformly at random from the training trajectories for poisoning.; ❷ RTG-only Selection, where only the selection mechanism based on the RTG metric is retained; and ❸ TD-only Selection, where only the selection mechanism based on the TD-error metric is retained. As shown in Fig. 5, the ASR for all three variants remains significantly lower than that of SpecDRL during the early stages of training This shows that even in the initial training phase, SpecDRL can rapidly establish a robust correlation between the backdoor trigger and the target action.

**Efficiency Under Budget Constraints.** To demonstrate the data efficiency of our framework, we conducted a comparative analysis under strictly constrained poisoning budgets.

*Table 3.* Comparative evaluation of five DRL backdoor attacks against the SHINE defense framework on the Atari Breakout environment. ↓ indicates that a lower value represents better evasion performance, while ↑ indicates better evasion for the attacker.

| Attack Method | SDR ↓ | FLP ↓ | EER ↑ | Pre-ASR | Post-ASR ↑ | CRR ↓ | Delay ↑ |
|---|---|---|---|---|---|---|---|
| TrojDRL | 82.4% | 78.5% | 17.2% | 99.81% | 27.31% | 99.4% | 1.0× |
| BadRL | 76.8% | 65.2% | 24.5% | 99.09% | 19.8% | 96.7% | 1.4× |
| SleeperNets | 41.2% | 32.8% | 36.4% | 99.98% | 51.2% | 93.5% | 2.2× |
| Q-Incept | 53.6% | 39.4% | 47.1% | **100%** | 42.5% | **91.2%** | 1.9× |
| SpecDRL (Ours) | **32.5%** | **25.3%** | **68.6%** | **100%** | **61.4%** | 94.2% | **3.1×** |

**Note:** All experiments are conducted in the **Breakout-v4** environment.
**Metric Definitions:** SDR: State-level Detection Rate; FLP: Feature-level Precision; EER: Explanation Evasion Rate; Pre-ASR: Attack Success Rate before SHINE shielding; Post-ASR: Attack Success Rate after SHINE shielding; CRR: Clean Reward Recovery; Delay: Training convergence delay factor.

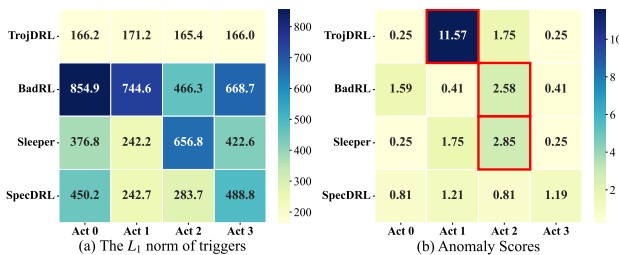
(a) The $L_1$ norm of triggers  (b) Anomaly Scores

*Figure 6.* Forensic Analysis. (a) The $L_1$ norm of triggers. (b) Anomaly Scores. The red boxes highlight the anomaly scores exceeding 2.

*Table 4.* Performance comparison under reduced poisoning budgets. Note that SpecDRL achieves superior ASR even with a significantly lower poisoning rate ($\beta = 0.01\%$) compared to the baselines ($\beta = 0.03\%$).

| Method Env | Baselines ($\beta = 0.03\%$) | | | **Ours ($\beta = 0.01\%$)** |
|---|---|---|---|---|
| | TrojDRL | BadRL | Sleeper | **SpecDRL** |
| *Atari Environments* | | | | |
| BeamRider (100k) | 11.08% | 21.20% | 7.92% | **68.47%** |
| Assault (100k) | 53.59% | 34.37% | 40.30% | **97.59%** |
| Gopher (100k) | 17.14% | 8.84% | 19.19% | **20.12%** |
| RoadRunner (700k) | 1.96% | 10.69% | 1.60% | **63.48%** |
| Atlantis (100k) | 28.65% | 25.66% | 26.09% | **30.12%** |
| Qbert (300k) | 6.24% | 5.69% | 3.70% | **24.31%** |
| Breakout (100k) | 55.15% | 41.11% | 62.69% | **99.75%** |

As presented in Table 4, we reduced the poisoning rate $\beta$ of SpecDRL to **0.01%**, which represents merely one-third of the budget allocated to the baseline methods ($\beta = 0.03\%$). Remarkably, despite this significant quantitative disadvantage, SpecDRL consistently outperforms all baselines across the evaluated environments.

## 6. Conclusions

This work establishes a new backdoor attack methodology for DRL. Through the integrated SpecDRL framework, we show that Subspace-Aware Injection, Value-Guided Strategic Sampling, and Bellman-Consistent Dynamic Reward Poisoning can jointly overcome the traditional efficiency–stealth trade-off. SpecDRL achieves high attack success

while preserving benign performance and maintaining theoretical consistency, providing new insights into DRL vulnerabilities.

Beyond introducing a powerful attack paradigm, this work also serves as a stress test for DRL systems by exposing weaknesses in value estimation and temporal credit assignment. Experimental results across Atari games, safety-oriented control tasks, and highway decision-making environments further suggest that these vulnerabilities are not limited to a specific observation modality or reward structure. Instead, they reveal broader security risks inherent in DRL optimization dynamics. We hope these findings will motivate the development of poisoning-aware training methods, trajectory-level consistency verification, and more robust policy regularization for safety-critical DRL deployments.

## Acknowledgements

This work was supported by Zhejiang Provincial Natural Science Foundation of China under Grant LQN25F010018, LQ24A010023 and LZ22F020007, and the National Key Research and Development Program of China under Grant 2024YFB3108100, and National Natural Science Foundation of China under Grants 62476250, 62506339, U23B2054, 62472335, 62372137, and Project of Zhejiang Provincial Department of Education FX2025047.

## Impact Statement

This paper studies backdoor attacks in deep reinforcement learning systems. The goal of this work is to better understand the security vulnerabilities of DRL agents and promote the development of more robust and trustworthy reinforcement learning systems. Although the proposed attack framework could potentially be misused, we believe that identifying and analyzing such threats is essential for designing effective defenses and improving the safety of future intelligent decision-making systems.

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

# A. Extended Preliminaries and Definitions

The reinforcement learning (RL) framework is grounded in the Markov Decision Process (MDP), which provides a mathematical abstraction for sequential decision-making. The tuple $M = \langle \mathcal{S}, \mathcal{A}, \mathcal{P}, \mathcal{R}, \gamma \rangle$ is defined as follows:

- State Space $\mathcal{S}$: A set of all possible environment configurations. In our experiments, this ranges from low-dimensional continuous vectors (SafetyGym) to high-dimensional pixel observations (Atari).

- Action Space $\mathcal{A}$: The set of all permissible actions. We consider both discrete action spaces $\mathcal{A} = \{a_1, \ldots, a_n\}$ and continuous action spaces $\mathcal{A} \subseteq \mathbb{R}^d$.

- Transition Probability $\mathcal{P}$: The dynamics function $\mathcal{P}(s_{t+1} \mid s_t, a_t)$, representing the probability of transitioning to $s_{t+1}$ given action $a_t$ in state $s_t$.

- Reward Function $\mathcal{R}$: A mapping $\mathcal{R} : \mathcal{S} \times \mathcal{A} \to \mathbb{R}$ that provides an immediate scalar feedback $r_t$ to the agent.

- Discount Factor $\gamma \in (0, 1)$: A scalar that determines the present value of future rewards, ensuring the convergence of infinite-horizon cumulative returns.

**Value Functions.** Given a stationary policy $\pi(a \mid s)$, the state-value function $V^\pi(s)$ and action-value function $Q^\pi(s, a)$ are defined as the expected discounted return:

$$V^\pi(s) = \mathbb{E}_\pi \left[ \sum_{t=0}^\infty \gamma^t r_t \,\middle|\, s_0 = s \right], \tag{17}$$

$$Q^\pi(s, a) = \mathbb{E}_\pi \left[ \sum_{t=0}^\infty \gamma^t r_t \,\middle|\, s_0 = s, a_0 = a \right]. \tag{18}$$

These functions satisfy the Bellman expectation equations:

$$V^\pi(s) = \sum_{a \in \mathcal{A}} \pi(a \mid s) \sum_{s' \in \mathcal{S}} \mathcal{P}(s' \mid s, a) \left[ r(s, a, s') + \gamma V^\pi(s') \right], \tag{19}$$

$$Q^\pi(s, a) = \sum_{s' \in \mathcal{S}} \mathcal{P}(s' \mid s, a) \left[ r(s, a, s') + \gamma \sum_{a' \in \mathcal{A}} \pi(a' \mid s') Q^\pi(s', a') \right]. \tag{20}$$

**Temporal Consistency.** For a given transition $(s_t, a_t, r_t, s_{t+1})$, the temporal-difference (TD) residual is defined as

$$\delta_t = r_t + \gamma V^\pi(s_{t+1}) - V^\pi(s_t), \tag{21}$$

which measures the degree of consistency between the observed reward and the value estimate under policy $\pi$. This quantity plays a central role in value-based learning and provides a natural signal for detecting anomalous or poisoned transitions.

# B. Algorithm

### B.1. Subspace-Aware Stealthy Trigger Injection

**Image-Space Trigger Construction.** In high-dimensional image-based environments, directly adding dense perturbations to raw pixel observations may introduce visible artifacts and compromise stealthiness. To address this issue, we design a spatially sparse trigger by exploiting pixel-level loading coefficients derived from the weakest-sensitive principal direction.

Let an image observation be represented as $s \in \mathbb{R}^{H \times W \times C}$, and let the corresponding flattened vector be $\text{vec}(s) \in \mathbb{R}^d$ with $d = H \cdot W \cdot C$. After performing PCA on benign flattened observations, we obtain the eigenvector $u_d \in \mathbb{R}^d$ associated with the smallest eigenvalue.

We reshape $u_d$ into a spatial tensor $\mathcal{R}(u_d) \in \mathbb{R}^{H \times W \times C}$. For each spatial location $(i, j)$, we define the loading coefficient $L_{i,j}$, which measures the sensitivity of pixel $(i, j)$ to perturbations along the weakest principal direction.

To restrict the trigger to low-saliency regions, we construct a binary injection mask $M \in \{0, 1\}^{H \times W}$ as

$$M_{i,j} = \mathbb{I}\left(L_{i,j} \leq \mathrm{percentile}(L, \theta)\right), \tag{22}$$

where $\mathrm{percentile}(L, \theta)$ denotes the $\theta$-th percentile of all loading coefficients, and $\mathbb{I}(\cdot)$ is the indicator function.

The final poisoned image is generated as

$$s^\dagger = s + \epsilon \cdot (M \odot \mathcal{R}(u_d)), \tag{23}$$

where $\odot$ denotes the Hadamard product. This construction ensures that perturbations are injected only into background regions with minimal semantic and perceptual importance, thereby achieving strong visual and statistical stealthiness.

**Algorithm: Subspace-Aware Trigger Generation.** Algorithm 1 formalizes how SpecDRL identifies the weakest-sensitive direction from benign state distributions and uses it as a stealthy trigger carrier for subsequent poisoning (described in Section 4.1).

---

**Algorithm 1** Subspace-Aware Trigger Generation

---

1: **Input:** Batch of benign states $X = \{x^{(i)}\}_{i=1}^{N}$, Perturbation magnitude $\epsilon$
2: **Output:** Trigger pattern vector $u_{trigger}$
3: *// Step 1: State Covariance Analysis*
4: Center the data: $\bar{x} \leftarrow \frac{1}{N} \sum_{i=1}^{N} x^{(i)}$
5: Compute covariance matrix: $\Sigma \leftarrow \frac{1}{N-1} \sum_{i=1}^{N} (x^{(i)} - \bar{x})(x^{(i)} - \bar{x})^T$
6: *// Step 2: Eigen Decomposition*
7: $\{u_1, \ldots, u_d\} \leftarrow \mathrm{EigenDecomposition}(\Sigma)$
8: Sort eigenvalues: $\lambda_1 \geq \cdots \geq \lambda_d$
9: *// Step 3: Weakest Sensitive Direction Selection*
10: Select eigenvector corresponding to $\lambda_{\min}$: $u_{trigger} \leftarrow u_d$
11: **return** Trigger direction $u_{trigger}$ for injection ($s^\dagger \leftarrow s + \epsilon \cdot u_{trigger}$)

---

### B.2. Value-Guided Sampling Details

Algorithm 2 provides implementation details for the value-guided strategic sampling procedure described in Section 4.2.

---

**Algorithm 2** Value-Guided Strategic Sampling

---

1: **Input:** Trajectory $\tau$ of length $L$, Poisoning rate $\beta$, Relaxation coefficient $\eta$, Value network $V_\phi$
2: **Output:** Set of poisoning indices $T_{poison}$
3: Calculate budget: $K \leftarrow \lfloor \beta \cdot L \rfloor$
4: Initialize candidate pool $C_{cand} \leftarrow \emptyset$
5: *// Phase 1: High-Value Filtering via RTG*
6: **for** $t \leftarrow 0$ **to** $L - 1$ **do**
7:     Calculate Return-to-Go: $G_t \leftarrow \sum_{k=0}^{\infty} \gamma^k r_{t+k}$
8: **end for**
9: Select top $m = \lfloor \eta \cdot K \rfloor$ indices by $G_t$:
10: $C_{cand} \leftarrow \mathrm{Top}\text{-}m(\{G_t\}_{t=0}^{L-1})$
11: *// Phase 2: Stability Pruning via TD-error*
12: **for all** $t \in C_{cand}$ **do**
13:     Calculate TD-error: $\delta_t \leftarrow r_t + \gamma V_\phi(s_{t+1}) - V_\phi(s_t)$
14: **end for**
15: Select bottom $K$ indices by absolute TD-error $|\delta_t|$:
16: $T_{poison} \leftarrow \mathrm{Bottom}\text{-}K(\{|\delta_t| \mid t \in C_{cand}\})$
17: **return** Optimal poisoning indices $T_{poison}$

---

### B.3. Bellman-Consistent Dynamic Reward Poisoning

Algorithm 3 provides implementation details for the bellman-consistent dynamic reward poisoning described in Section 4.3.

---

**Algorithm 3** Bellman-Consistent Dynamic Reward Poisoning

---

1: **Input:** Original trajectory $\tau$, Poisoning indices $T_{poison}$, Target action $a^\dagger$, Trigger $u_d$
2: **Output:** Poisoned trajectory $\tau^\dagger$
3: Initialize $\tau^\dagger \leftarrow \tau$
4: **for** $t \in T_{poison}$ **do**
5:     *// State Injection*
6:     $s_t^\dagger \leftarrow s_t + \epsilon \cdot u_d$; Update $s_t$ in $\tau^\dagger$
7:     *// Dynamic Reward Formulation*
8:     Calculate Induction Kernel: $\mathcal{J} \leftarrow c(2 \cdot \mathbb{I}\{a_t = a^\dagger\} - 1)$
9:     Calculate Suppression Term: $\mathcal{R}_{future} \leftarrow G_t - r_t$
10:     Solve Adaptive Coefficient $\alpha_t$ (via Theorem C.3)
11:     Store original reward: $r_{old} \leftarrow r_t$
12:     Compute poisoned reward: $r_t^\dagger \leftarrow \mathcal{J} - \alpha_t \cdot \mathcal{R}_{future}$
13:     Update $r_t$ in $\tau^\dagger$ to $r_t^\dagger$
14:     *// Differential Temporal Backfilling*
15:     **if** $t > 0$ **then**
16:         Calculate correction: $r_{t-1}^\dagger \leftarrow r_{t-1} + \gamma(r_{old} - r_t^\dagger)$
17:         Update $r_{t-1}$ in $\tau^\dagger$ to $r_{t-1}^\dagger$
18:     **end if**
19: **end for**
20: **return** Poisoned trajectory $\tau^\dagger$ for agent update

---

# C. Theoretical Analysis and Proof

## C.1. Attack Effectiveness Guarantee

In this section, we aim to rigorously demonstrate the core mechanism of SpecDRL from a theoretical perspective. To prove that our dual filtering criteria—namely, high Return-to-Go (RTG) and low Temporal Difference (TD) error—can maximize attack efficacy under limited poisoning budgets, we first establish a sensitivity analysis model of the value function with respect to reward changes. Finally, we provide theoretical guarantees for the convergence of the policy towards the target action.

### C.1.1. LINEARIZED REPRESENTATION OF VALUE PROPAGATION

First, we consider the background in which the attack occurs: the attacker samples and poisons data under a relatively stable policy $\pi$. Under this fixed policy $\pi$, the original complex Markov Decision Process (MDP) degenerates into a simple Markov Reward Process (MRP), where state transitions depend solely on environmental dynamics. At this point, the state value function $V(s)$, which describes the expected cumulative return starting from state $s$, must satisfy the classical Bellman equation:

$$V(s) = r(s) + \gamma \sum_{s' \in \mathcal{S}} P(s'|s)V(s') \tag{24}$$

To analyze how local reward modifications diffuse into global value estimates, we rewrite the above recursive relationship into a matrix form that is easier to analyze. Let $V$ and $r$ be the value vector and reward vector for all states, respectively, and let $P$ be the state transition matrix. By rearranging terms and inverting the matrix, we can explicitly solve for the value vector:

$$V = r + \gamma PV \implies V = (I - \gamma P)^{-1} r \tag{25}$$

The inverse matrix $(I - \gamma P)^{-1}$ carries profound physical significance. Since the discount factor $\gamma \in (0, 1)$ and $P$ is a stochastic matrix, this inverse can be expanded via the Neumann Series as $I + \gamma P + \gamma^2 P^2 + \dots$. This essentially represents the Successor Representation or the discounted visitation distribution of successor states. It encodes the discounted frequency of reaching all future states from the current state, implying that a reward signal $r$ at any single point propagates to the entire state space through this matrix.

Based on this, we can further derive the dependency of the action-value function $Q(s, a)$ on the reward. $Q(s, a)$ consists of

the immediate reward and the value of the next state. Using the matrix expression above, we can express $Q(s,a)$ as a linear function of the reward vector $r$:

$$Q(s,a) = \mathbf{e}_s^\top r + \gamma \sum_{s' \in \mathcal{S}} P(s'|s,a)\mathbf{e}_{s'}^\top (I - \gamma P)^{-1} r = \mathbf{f}_a(s)^\top r \tag{26}$$

Here, we encapsulate the complex matrix operations into a row vector $\mathbf{f}_a(s)^\top$. This vector is not merely a mathematical coefficient; fundamentally, it defines the **Linear Sensitivity** of the value function. It quantifies the magnitude of change in $Q(s,a)$ resulting from a reward modification at any arbitrary time step. This linear relationship forms the basis for the feasibility of the attack:

$$\Delta Q(s,a) = \mathbf{f}_a(s)^\top \Delta r \tag{27}$$

### C.1.2. MATHEMATICAL FORMALIZATION OF ATTACK OBJECTIVES

The attacker's ultimate goal is to manipulate the agent's selection, ensuring it firmly chooses the target action $a^\dagger$ at the trigger state $s^\dagger$. To quantify the degree to which this goal is achieved, we define the Action Value Margin $M$. This variable represents the gap between the Q-value of the target action $a^\dagger$ and that of the current optimal non-target action:

$$M := Q^\pi(s^\dagger, a^\dagger) - \max_{a \neq a^\dagger} Q^\pi(s^\dagger, a) \tag{28}$$

Clearly, the attack is successful only when $M > 0$. Therefore, the attacker's task is to maximize the increment of this margin, $\Delta M$, by applying a reward perturbation $\Delta r$. Using the sensitivity formula derived previously, we can directly link $\Delta M$ to the perturbation $\Delta r$:

$$\Delta M = \mathbf{w}^\top \Delta r \tag{29}$$

where we define the vector

$$\mathbf{w} := \mathbf{f}_{a^\dagger}(s^\dagger) - \mathbf{f}_{a^-}(s^\dagger), \tag{30}$$

with

$$a^- := \arg \max_{a \in \mathcal{A} \setminus \{a^\dagger\}} Q^\pi(s^\dagger, a), \tag{31}$$

as the Sensitivity Difference Vector. Each element $w_i$ in $\mathbf{w}$ precisely quantifies the contribution of modifying the reward at the $i$-th time step towards widening the gap between the target and non-target actions. This serves as the theoretical root for our focus on High RTG (i.e., high gradient contribution) in the sampling strategy. This formulation provides the theoretical justification for our focus on High RTG in the sampling strategy, which is grounded in the Policy Gradient Theorem (Sutton & Barto, 2018). The theorem states that the gradient of the performance objective is proportional to the advantage function: $\nabla J(\theta) \propto \mathbb{E}[A^\pi(s,a)\nabla \log \pi_\theta(a|s)]$. Since the Return-to-Go (RTG) $G_t$ serves as a Monte Carlo estimate of the action value (and by extension, the advantage), selecting time steps with high RTG is equivalent to targeting the dimensions with the largest scalar coefficients in the gradient update. Consequently, maximizing the margin $\Delta M$ at these moments does not merely increase the Q-value numerically, but aligns the reward perturbation $\Delta r$ with the direction of steepest ascent for the target action's probability $\pi_\theta(a^\dagger|s^\dagger)$.

### C.1.3. ADVERSARIAL CONSTRAINTS AND OPTIMIZATION SOLUTION

However, merely pursuing the maximization of $\Delta M$ is prone to failure. This is because reinforcement learning is an online update process, and the agent possesses "self-correction" capabilities. When we modify the reward at time step $i$, we inevitably alter the TD-error $\delta_i$ at that moment. The agent subsequently adjusts its value estimate according to the TD update rule $V(s) \leftarrow V(s) + \eta_v \delta_i$. This update generates a Cancellation Effect: the agent attempts to adapt to the modified reward by adjusting $V$, thus digesting the signal injected by the attacker.

Therefore, a prudent attacker must find a balance between maximizing the gain of the attack and minimizing the risk of agent correction. We introduce this constraint into the optimization objective: under the premises of the poisoning magnitude constraint $\|\Delta r\|_\infty \leq C$ and the sparsity constraint $\|\Delta r\|_0 \leq k$, we must not only maximize the margin increase, but also penalize potential corrections caused by high TD errors. We model this as the following optimization problem:

$$\max_{\Delta r} \mathbf{w}^\top \Delta r - \lambda \sum_i c_i |\delta_i| |\Delta r_i|$$
$$\text{s.t. } \|\Delta r\|_\infty \leq C, \quad \|\Delta r\|_0 \leq k \tag{32}$$

where $\lambda$ is a trade-off coefficient. We define the term $q_i = |w_i| - \lambda c_i |\delta_i|$ as the Effective Poisoning Gain. Observing this objective function reveals the optimal strategy: to maximize the total objective, we should select the $k$ time steps with the largest $q_i$ values for full-magnitude injection. This mathematical conclusion perfectly corresponds to the intuition of SpecDRL: the optimal timing for poisoning is exactly those moments that make a large contribution to the final objective (high $|w_i|$ means high RTG) while the agent's current cognition is stable (low $|\delta_i|$ means less likely to trigger drastic corrections).

### C.1.4. CONVERGENCE THEOREM AND PROOF

Based on the derivation above, we can now present the core theorem regarding the final effect of the attack, guaranteeing that under our strategy, the model parameters will indeed converge towards the target direction.

**Theorem C.1** (Targeted Policy Induction). *If the trigger state $s^\dagger$ is selected via the criteria of maximum RTG and minimum TD-error, then after the training process converges, there exists a parameter point $\theta^*$ satisfying:*

$$\pi_{\theta^*}(a^\dagger | s^\dagger) = \max_a \pi_{\theta^*}(a | s^\dagger) \tag{33}$$

*Proof.* Let $M(\pi) = Q^\pi(s^\dagger, a^\dagger) - \max_{a \neq a^\dagger} Q^\pi(s^\dagger, a)$ be the action value margin.

The sampling strategy identifies the poisoning set $T_{poison}$ (as defined in Section 4.2) which maximizes the cumulative effective gain. By the linear sensitivity derived in Eq. (21), the resulting reward perturbation $\Delta r^*$ satisfies:

$$\Delta M = \mathbf{w}^\top \Delta r^* = \sum_{t \in T_{poison}} w_t C \geq \mathbf{w}^\top \Delta r, \quad \forall \|\Delta r\|_0 \leq k \tag{34}$$

Since the benign margin is bounded, a sufficiently large $C$ guarantees $Q^\pi(s^\dagger, a^\dagger) \gg Q^\pi(s^\dagger, a)$ for all $a \neq a^\dagger$.

The policy gradient direction at $s^\dagger$ is dominated by the advantage function:

$$\mathbb{E}[\nabla_\theta J] \propto \mathbb{E}[A^\pi(s^\dagger, a^\dagger) \nabla \ln \pi(a^\dagger | s^\dagger)] > 0 \tag{35}$$

The constraint $\min |\delta_t|$ minimizes the variance of the value estimate $\mathrm{Var}(A^\pi)$, ensuring the gradient update direction is strictly consistent with the margin maximization.

At convergence $\theta^*$, the ordering of Q-values is preserved by the monotonic Softmax operator:

$$Q^{\pi_{\theta^*}}(s^\dagger, a^\dagger) > \max_{a \neq a^\dagger} Q^{\pi_{\theta^*}}(s^\dagger, a) \implies \pi_{\theta^*}(a^\dagger \mid s^\dagger) > \max_{a \neq a^\dagger} \pi_{\theta^*}(a \mid s^\dagger) \tag{36}$$

Therefore, $\pi_{\theta^*}(a^\dagger \mid s^\dagger) = \max_a \pi_{\theta^*}(a \mid s^\dagger)$. $\qquad\square$

## C.2. Analysis of Utility Preservation on Benign Tasks

Having established the theoretical guarantees for attack effectiveness in the previous section, we now turn our attention to the impact of the attack on the agent's performance in benign tasks. A stealthy adversarial attack must satisfy a dual objective: maximizing the target margin $M$ at the trigger state (attack success) while minimizing the deviation in the global episodic return $G_0$ (utility preservation). This ensures that the agent's general behavior remains functional and the attack remains imperceptible during normal operation.

In this section, we first analyze the upper bound of return bias under standard sparse constraints, revealing the inherent limitations of naive poisoning methods. Subsequently, we provide a theoretical proof demonstrating how the "Dynamic Differential Backfilling" mechanism proposed in SpecDRL guarantees return invariance.

### C.2.1. BIAS UPPER BOUND UNDER SPARSE CONSTRAINTS

Consider a finite trajectory $\tau = \{s_0, a_0, r_0, \ldots, s_{L-1}, a_{L-1}, r_{L-1}\}$ of length $L$. The discounted return starting from time step $t$ is denoted as $G_t(r)$, and specifically, the return for the entire episode is $G_0(r)$.

In a standard poisoning setting, the reward perturbation can be modeled as an additive vector $\Delta r$. This perturbation is subject to a magnitude constraint $\|\Delta r\|_\infty \leq C$ and a sparsity constraint $\|\Delta r\|_0 \leq k = \beta L$, where $\beta \ll 1$ represents the

poisoning rate. Under these conditions, the variation in the episodic return, denoted as $\Delta G_0$, is:

$$\Delta G_0 := G_0(r^\dagger) - G_0(r) = \sum_{t=0}^{L-1} \gamma^t \Delta r_t \tag{37}$$

By applying the triangle inequality, we derive the worst-case upper bound:

$$\begin{aligned}
|\Delta G_0| &\leq \sum_{t:\Delta r_t \neq 0} \gamma^t |\Delta r_t| \\
&\leq C \sum_{t:\Delta r_t \neq 0} \gamma^t \\
&\leq C \|\Delta r\|_0 \\
&\leq C\beta L
\end{aligned} \tag{38}$$

The derivation above indicates that under general sparse reward poisoning, the worst-case upper bound of the episodic return deviation grows linearly with the poisoning rate $\beta$ and the trajectory length $L$. Particularly in Deep Reinforcement Learning scenarios where an episode may contain thousands of steps, even a minimal $\beta$ can result in a significant cumulative drop in return.

From the attacker's perspective, this implies that uncontrolled poisoning inevitably degrades benign task performance. As the policy updates over repeated sampling, this accumulated bias manifests as noticeable performance degradation during the testing phase, potentially exposing the backdoor attack.

### C.2.2. RETURN INVARIANCE VIA DIFFERENTIAL BACKFILLING

To mitigate the aforementioned performance degradation, SpecDRL introduces a Dynamic Differential Backfilling mechanism. Unlike standard attacks that strictly inject positive rewards, this mechanism compensates for the value injection by modifying the reward of the preceding time step. We present the following theorem to demonstrate that this mechanism strictly bounds the return deviation within a constant range, independent of the trajectory length or attack frequency.

**Theorem C.2** (Return Preservation Property). *For any trajectory $\tau$ and its original reward sequence $r$, if the poisoned reward sequence $r^\dagger$ is generated via the Dynamic Differential Backfilling rule, the absolute deviation in episodic return is bounded by the single-step constraint:*

$$|G_0(r^\dagger) - G_0(r)| \leq C \tag{39}$$

*Proof.* Let $T_{poison}$ be the set of time steps selected for poisoning based on the criteria in Section 5.1. The SpecDRL mechanism modifies the rewards as follows:

For any target step $t \in T_{poison}$, the current reward is modified as:

$$r_t^\dagger = r_t + \Delta r_t, \quad |\Delta r_t| \leq C \tag{40}$$

If $t > 0$, the mechanism propagates a negative compensation to the immediate predecessor:

$$r_{t-1}^\dagger = r_{t-1} + \gamma(r_t - r_t^\dagger) = r_{t-1} - \gamma \Delta r_t \tag{41}$$

Rewards for all other time steps remain unchanged. We can define the effective perturbation $\Delta r_k$ for any step $k$. By convention, let $\Delta r_L = 0$. The modified reward sequence can be unified as:

$$r_k^\dagger = r_k + \Delta r_k - \gamma \Delta r_{k+1} \tag{42}$$

where $\Delta r_k \neq 0$ only if $k \in T_{poison}$. Substituting this into the episodic return difference:

$$\begin{aligned}
|G_0(r^\dagger) - G_0(r)| &= \left| \sum_{k=0}^{L-1} \gamma^k (r_k^\dagger - r_k) \right| \\
&= \left| \sum_{k=0}^{L-1} \gamma^k (\Delta r_k - \gamma \Delta r_{k+1}) \right| \\
&= \left| \sum_{k=0}^{L-1} \gamma^k \Delta r_k - \sum_{k=0}^{L-1} \gamma^{k+1} \Delta r_{k+1} \right|
\end{aligned} \tag{43}$$

Observing the summation structure, this forms a telescoping sum. The terms perfectly cancel out pairwise. Expanding the series yields:

$$
\begin{aligned}
&= \left| |\Delta r_0| + \sum_{i=1}^{L-1} \gamma^i \Delta r_i - \sum_{i=1}^{L-1} \gamma^i \Delta r_i - \gamma^L \Delta r_L \right| \\
&= |\Delta r_0| \\
&\leq C
\end{aligned}
\tag{44}
$$

We have effectively corrected the error introduced by each attack immediately along the time axis. This means that regardless of how many attacks are launched, the cumulative error does not accumulate but is strictly confined within a negligible constant range. This theoretically guarantees the high stealthiness of the attack—altering critical decisions without disrupting the overall performance on benign tasks. □

### C.2.3. BELLMAN CONSISTENCY OF THE VALUE FUNCTION

Theorem C.2 establishes a critical theoretical foundation for utility preservation. By rigorously proving that the cumulative return deviation is bounded by a constant ($O(1)$), it effectively resolves the issue of linear error growth ($O(L)$) inherent in naive poisoning methods. This ensures that, from a macroscopic perspective, the agent's long-term performance metric remains stable regardless of the task horizon.

Building upon this global guarantee, we further advance our theoretical framework to the *micro-level* (single-step transition) to achieve finer-grained control over the learning dynamics. While global return invariance constrains the trajectory sum, the practical optimization of DRL agents is driven by local Temporal-Difference (TD) signals at each time step. Extending our analysis to this granular level offers two significant contributions:

1. **Stabilizing Optimization Dynamics:** By enforcing consistency at individual time steps, we minimize the variance of local gradient estimates. This prevents potential oscillations in the loss function during backpropagation, ensuring that the injection of the backdoor does not disrupt the convergence stability of the policy network.

2. **Structural Alignment via Bellman Operator:** Micro-level analysis ensures that the poisoned value function is not just numerically equivalent in the long run, but *structurally consistent* under the Bellman operator. This implies that the geometric landscape of the poisoned value function $V(s^\dagger)$ closely aligns with the benign manifold $V(s)$. Consequently, the distributional properties of the TD-error remain invariant, eliminating local statistical anomalies that could be exploited by fine-grained detection mechanisms.

In this section, we prove that the introduction of the dynamic scaling coefficient $\alpha_t$ satisfies this rigorous requirement, enforcing strict $\epsilon$-consistency between the poisoned and original value functions.

**Theorem C.3** (Bellman Consistency via Dynamic Scaling). *Under a fixed policy $\pi$, for any given tolerance $\epsilon > 0$, there exists a dynamic scaling coefficient $\alpha_t$ such that the discrepancy between the value function of the trigger state $s^\dagger$ and the original state $s$ satisfies:*

$$
|V(s^\dagger) - V(s)| \leq \epsilon
\tag{45}
$$

*Proof.* Under a fixed policy $\pi$, the MDP degenerates into a Markov Reward Process (MRP). Recalling the standard Bellman equation defined in Eq. (22), the value function $V(s)$ satisfies the recursive relationship.

For the trigger state $s^\dagger$, SpecDRL modifies only the immediate reward function while strictly preserving the environmental dynamics (i.e., $P(s'|s^\dagger) = P(s'|s)$). This isolation of the reward term allows us to simplify the value discrepancy. Since the transition probabilities are identical, the expected future values cancel out perfectly, reducing the discrepancy to the difference in immediate rewards:

$$
\begin{aligned}
V(s^\dagger) - V(s) &= r^\dagger(s^\dagger) - r(s) + \gamma \sum_{s' \in \mathcal{S}} \underbrace{\left( P(s'|s^\dagger) - P(s'|s) \right)}_{=0} V(s') \\
&= r^\dagger(s^\dagger) - r(s)
\end{aligned}
\tag{46}
$$

Substituting the dynamic reward reformulation defined in Section 4.3, the poisoned reward is constructed as:

$$
r^\dagger(s^\dagger) - r(s) = \mathcal{J}(a_t, a^\dagger) - \alpha_t \cdot \mathcal{R}_{future}(t) - r(s)
\tag{47}
$$

From the definition of cumulative return, the expected value of the suppression term $\mathcal{R}_{future}(t)$ precisely corresponds to the future value component of the Bellman equation:

$$
\begin{aligned}
\mathbb{E}\left[V(s^\dagger) - V(s) \mid S_t = s\right] &= r^\dagger(s^\dagger) - r(s) \\
&= \mathcal{J}(a_t, a^\dagger) - \alpha_t \underbrace{\mathbb{E}\left[\mathcal{R}_{future}(t)\right]}_{V(s)-r(s)} - r(s) \\
&= \mathcal{J}(a_t, a^\dagger) - \alpha_t(V(s) - r(s)) - r(s)
\end{aligned}
\tag{48}
$$

To enforce strict value consistency, we set the expected discrepancy to zero:

$$
\begin{aligned}
\mathbb{E}[V(s^\dagger) - V(s) \mid S_t = s] &= \mathcal{J}(a_t, a^\dagger) - \alpha_t(V(s) - r(s)) - r(s) \\
&= 0
\end{aligned}
\tag{49}
$$

Solving for $\alpha_t$, we obtain the closed-form solution:

$$
\alpha_t = \frac{\mathcal{J}(a_t, a^\dagger) - r(s)}{V(s) - r(s)}
\tag{50}
$$

This result demonstrates that for any given state, there exists a unique scalar $\alpha_t$ that perfectly balances the induction kernel $\mathcal{J}$ against the future value expectation. By dynamically adjusting $\alpha_t$ at each step, the suppression term effectively neutralizes the deviation, ensuring that $|V(s^\dagger) - V(s)| \leq \epsilon$ holds universally. In practice, we compute $\alpha_t$ using a Monte Carlo estimate of the trajectory return, such as $G_t$, rather than the exact state value $V_t$. This provides a tractable, online-compatible approximation while still preserving the intended behavior of dynamic reward scaling.

$\square$

# D. Experimental Details

## D.1. Computational Resources

All experiments presented in this paper were conducted on a dedicated Dell Precision 7960 Tower workstation. The computational environment is equipped with an Intel Xeon w5-3435X processor and an NVIDIA RTX A6000 GPU with 48GB of VRAM. To support the extensive memory requirements of training replay buffers, the system is provisioned with 256GB of DDR5 ECC RAM. The operating system used was Ubuntu 22.04 LTS, with deep learning frameworks implemented in PyTorch utilizing CUDA acceleration.

## D.2. Baselines

To comprehensively evaluate the performance boundaries of SpecDRL, we systematically compared it with the SOTA methods representing different attack paradigms:

- **TrojDRL:** Adopts a traditional periodic injection strategy. It does not rely on state filtering but triggers indiscriminately based on preset time intervals and a fixed budget.

- **BadRL:** A value-based method that utilizes an externally trained Q-network to assess state importance. It filters high-score candidates for precision attacks but lacks consideration for reward dynamics.

- **SleeperNets:** Employs a random time-step sampling strategy and introduces a reward rewriting mechanism. This method represents the mainstream level of current outer-loop attacks.

To eliminate bias caused by implementation details, all baseline methods were implemented under a unified state-reward poisoning interface and evaluated fairly under identical environmental configurations and total poisoning budgets.

## D.3. Implementation of Subspace-Aware Trigger Injection

To ensure the stealthiness of the backdoor trigger, SpecDRL employs a spectral injection method based on Principal Component Analysis (PCA), as illustrated in Figure 7. Unlike traditional methods that employ spatially localized pixel

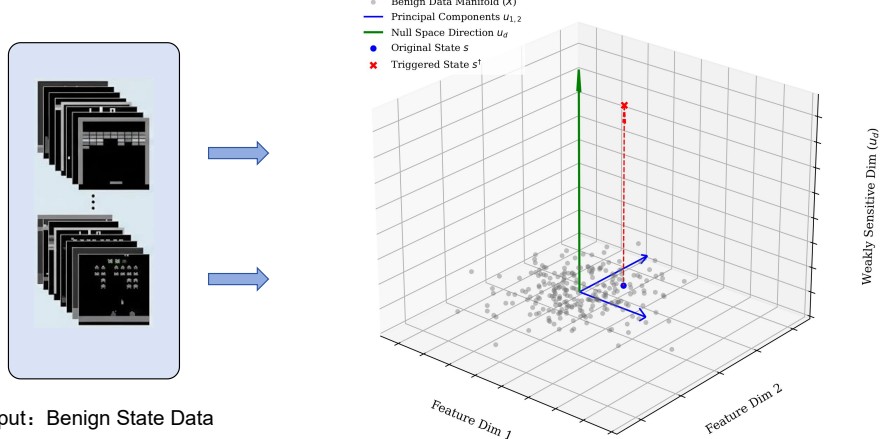

*Figure 7.* The framework of Spectral Subspace Analysis. SpecDRL utilizes PCA to identify the least influential dimension ($\lambda_{min}$) of the state space, providing a mathematical blind spot for stealthy trigger injection that is orthogonal to the task-relevant manifold.

patches (which introduce significant high-frequency artifacts), our approach identifies the "blind spots" of the state space—dimensions with minimal statistical variance that carry negligible information regarding environmental dynamics.

Specifically, the trigger generation process proceeds in three stages:

1. **Covariance Analysis:** Given a batch of $N$ benign states sampled from the environment, $S = \{s^{(1)}, \ldots, s^{(N)}\} \subset \mathbb{R}^d$, we first center the data and compute the empirical covariance matrix $\Sigma$:

$$\Sigma = \frac{1}{N-1} \sum_{i=1}^{N} (s^{(i)} - \bar{s})(s^{(i)} - \bar{s})^T \tag{51}$$

2. **Blind Spot Identification:** We perform eigenvalue decomposition on $\Sigma$ to obtain the orthogonal basis $U$ and eigenvalues $\Lambda$. We identify the eigenvector $u_d$ corresponding to the minimum eigenvalue $\lambda_{\min}$ (i.e., the direction of least variance). This vector $u_d$ represents a subspace orthogonal to the primary task manifold.

3. **Orthogonal Injection:** The trigger is injected as a global additive perturbation along this specific direction:

$$s^{\dagger} = s + \epsilon \cdot u_d \tag{52}$$

where $\epsilon$ is a scaling factor controlling the perturbation magnitude. Since $\lambda_{\min} \to 0$, variations along $u_d$ are typically filtered out by the agent's feature extractor as noise during normal training. However, by consistently associating this specific noise pattern with high rewards (via our poisoning strategy), the agent learns to recognize this "invisible" signal as a high-value feature, establishing a robust backdoor link.

### D.4. Extra Ablation

To further investigate the data efficiency and robustness of SpecDRL, we conducted an additional ablation study focusing on the impact of the poisoning rate $\beta$. While our primary experiments utilized a standard poisoning budget, real-world attacks often necessitate minimizing traces to the absolute limit to evade detection. We benchmarked the performance of SpecDRL against three baseline methods (TrojDRL, BadRL, and SleeperNets) operating under a fixed budget of $\beta = 0.03\%$. As summarized in the experimental results, SpecDRL demonstrates remarkable resilience in a subset of Atari environments, maintaining high performance even when the budget is reduced from 0.03% to 0.01%. In other environments, our method

*Table 5.* Performance comparison under reduced poisoning budgets. Note that **SpecDRL** maintains high ASR even with a significantly lower poisoning rate ($\beta = 0.02\%$) compared to the baselines ($\beta = 0.03\%$).

| Method / Env | Baselines ($\beta = 0.03\%$) | | | Ours ($\beta = 0.02\%$) |
|---|---|---|---|---|
| | TrojDRL | BadRL | Sleeper | **SpecDRL** |
| *Atari Environments* | | | | |
| Pong (300k) | 20.20% | 13.80% | 32.16% | **40.29%** |
| SpaceInvaders (200k) | 15.53% | 5.62% | 27.07% | **42.14%** |
| Jamesbond (350k) | 7.43% | 1.73% | 14.53% | **31.54%** |

*Table 6.* Sensitivity analysis of the reward shaping constant $c$. We evaluated the Attack Success Rate (ASR) of **SpecDRL** across five representative environments under varying $c$ values ($c \in \{0.1, 1.0, 2.0, 5.0, 10.0\}$). The results assist in determining the optimal hyperparameter range for ensuring stable backdoor injection. Best results are marked in **bold**.

| Environment | Reward Shaping Constant $c$ | | | | |
|---|---|---|---|---|---|
| | $c = 0.1$ | $c = 1.0$ | $c = 2.0$ | $c = 5.0$ | $c = 10.0$ |
| Breakout (100k) | 61.05% | 99.67% | 99.65% | **99.99%** | **99.99%** |
| Assault (100k) | 18.16% | 91.54% | 93.26% | 99.69% | **99.99%** |
| Atlantis (100k) | 40.56% | 34.09% | 73.09% | 84.87% | **95.90%** |
| Gopher (100k) | 7.01% | 5.10% | 14.87% | **57.47%** | 47.43% |
| BeamRider (100k) | 10.63% | 21.94% | 66.61% | **99.94%** | 0.02% |

sustains an ASR comparable to the standard setting even when $\beta$ is lowered to 0.02%. Notably, in the *Breakout* environment, even when the poisoning rate is reduced to one-tenth of the original magnitude ($\beta = 0.003\%$), the ASR during the early training stages remains substantial.

### D.5. Sensitivity Analysis on Reward Shaping Constant $c$

To determine the optimal magnitude for the reward shaping constant $c$, we conducted a fine-grained sensitivity analysis across five representative environments. The goal was to identify a hyperparameter range that balances the strength of the backdoor signal against the stability of the training process. The empirical results are detailed in Table 6.

**Impact of Signal Strength.** As observed in the lower range of the spectrum ($c \in \{0.1, 1.0\}$), the Attack Success Rate (ASR) is generally suboptimal. For instance, in complex environments like *BeamRider* and *Gopher*, a small $c$ fails to provide a sufficient gradient to override the benign reward signals, resulting in ASRs below 22%. This indicates that the backdoor incentive must meet a minimum threshold to be effectively learned by the agent.

**Stability Trade-off at High Values.** Increasing $c$ to 2.0 and 5.0 yields a significant performance boost, with *SpecDRL* achieving near-perfect attack success ($> 99\%$) in *Breakout*, *Assault*, and *BeamRider* at $c = 5.0$. However, the results at $c = 10.0$ reveal a critical stability boundary. While *Atlantis* benefits from the stronger signal (reaching 95.90%), *BeamRider* suffers a catastrophic failure, with ASR dropping to 0.02%. This collapse suggests that an excessively large $c$ can destabilize the value function approximation, likely causing the agent's policy to diverge or fall into a degenerate solution.

**Parameter Selection.** Based on these findings, we observe that $c = 5.0$ offers the most robust performance across diverse environments, providing strong enough incentives for effective backdoor injection while avoiding the instability associated with extreme reward scaling. Consequently, we prioritized the range of $c = 5.0$ for our main experiments to ensure consistent attack efficacy.

### D.6. Extended Experimental Results on Attack Performance

In the main text, we presented ASR learning curves for only a subset of representative environments due to space constraints. In this section, we provide additional experimental results to further validate the universality and robustness of **SpecDRL**. We extended our evaluation to cover a diverse range of Atari games, including *Pong*, *SpaceInvaders*, *Jamesbond*, *Gopher*, *Atlantis*, and *RoadRunner*. Figure 8 illustrates the Attack Success Rate (ASR) trajectories throughout the entire training process. As evidenced by these additional learning curves, **SpecDRL** consistently outperforms baseline methods (TrojDRL, BadRL, and SleeperNets) across all tested environments. A recurring pattern observed is that baseline methods frequently

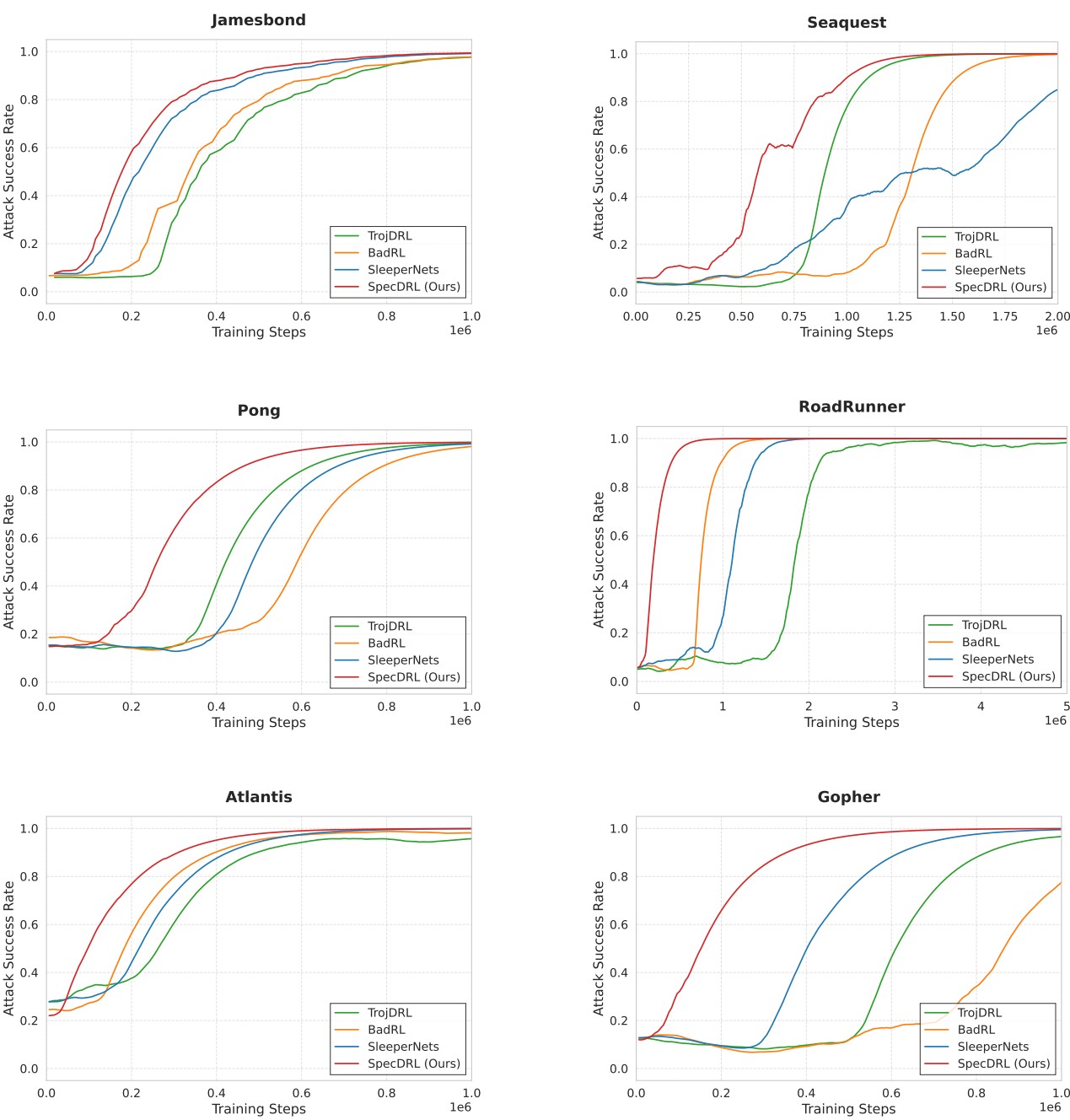

*Figure 8.* Additional learning curves of ASR across six supplementary Atari environments. **SpecDRL** consistently maintains superior induction efficiency across various task dynamics.

suffer from ASR instability and slow convergence, often requiring millions of additional training steps to reach comparable ASR levels. In contrast, **SpecDRL** typically saturates near 100% ASR within the first few hundred thousand steps. These comprehensive results reinforce our claim that the proposed subspace-aware trigger generation and value-guided injection mechanisms are highly effective and independent of specific game dynamics.

