# OpenReview forum: "Toward Subspace-Perturbed Trajectory-Aware Backdoor Attacks in Deep Reinforcement Learning"
_ICML.cc/2026/Conference — ICML 2026 regular_

### Official Review · Reviewer_swgT · 2026-03-11

**Soundness:** 3
**Presentation:** 4
**Significance:** 3
**Originality:** 3
**Overall Recommendation:** 4
**Confidence:** 2

**Summary:**

This paper introduces SpecDRL, a backdoor attack framework for deep reinforcement learning (DRL). The proposed framework consists of three core components: subspace-aware injection, value-guided strategic sampling, and Bellman-consistent dynamic reward poisoning. Specifically, the subspace-aware injection exploits perceptual blind spots by embedding triggers into the least sensitive subspaces of the state manifold. The value-guided sampling selects the most influential time steps for poisoning based on return-to-go and temporal-difference errors. Finally, the dynamic reward poisoning maintains reward integrity by analytically enforcing $\epsilon$-consistency of value functions and bounding global return deviations.

**Compliance With Llm Reviewing Policy:**

Affirmed.

**Final Justification:**

The authors have addressed my primary concerns regarding the computational complexity and the soundness of evaluation. I will keep my positive score.

**Key Questions For Authors:**

1.	What is the computational complexity and operational overhead of the proposed SpecDRL framework?
2.	Could the authors discuss potential ideas or defense strategies that might successfully mitigate this specific attack?

**Limitations:**

The paper currently lacks an Impact Statement, which is highly recommended for research introducing new adversarial attack methodologies.

**Strengths And Weaknesses:**

Strengths:

1.	Robustness against backdoor attacks is a critical issue in DRL. The proposed method makes a valuable contribution that will likely stimulate further research into backdoor defenses.
2.	This paper is well-written, clearly organized, and easy to follow.
3.	The majority of the core claims are well-supported by theoretical analysis or empirical experiments.
4.	To the best of my knowledge, the proposed SpecDRL framework introduces a novel methodology to the field.

Weaknesses:

1.	The current experiments focus on Atari games, which have pixel-based observations. Expanding the evaluation to benchmarks with vector-based observations, such as continuous control tasks in MuJoCo, would significantly strengthen the empirical soundness of the paper.

---

> ### Author Rebuttal · Authors · 2026-03-31
>
> We sincerely thank the reviewer for the encouraging and thoughtful evaluation of our work. We are especially encouraged that the reviewer considers SpecDRL a novel methodological contribution to the field. Our detailed responses are as follows:
>
> **Q1: Expanding the evaluation to benchmarks with vector-based observations.**
>
> **R1:** We added vector-based continuous-control experiments on Safety and Highway environments, and also included Q-Incept [1] (ICML 2025) as a stronger recent baseline in these new settings. A compact summary is shown below (all numbers are percentages):
>
> | Env.       | Q-Incept(ASR) | Q-Incept(BPI) | SpecDRL(ASR) | SpecDRL(BPI) |
> |------------|---------------|---------------|--------------|--------------|
> | CarGoal1   | 100           | 99.77         | 100          | 100          |
> | PointPush1 | 100           | 78.08         | 100          | 81.92        |
> | merge      | 31.24         | 96.82         | 39.24        | 100          |
>
> These results show that SpecDRL is not limited to pixel-based Atari inputs: on the added Safety tasks, it matches Q-Incept in ASR while achieving higher BPI, and on Highway-merge it outperforms Q-Incept in both ASR and BPI. We will also revise broad phrases such as "diverse observation modalities and control dynamics" so that the wording matches the validated empirical scope. Full quantitative results are included in the anonymous supplementary material (Table 1. Cross_Env_Attack_Comparison)
>
> [1] Rathbun et al. Adversarial Inception for Bounded Backdoor Poisoning in Deep Reinforcement Learning. ICML 2025.
>
> **Q2: What is the computational complexity and operational overhead of the proposed SpecDRL framework?**
>
> **R2:** SpecDRL total 1.3M, identical to PPO, and its per-step online cost is 0.0205 GFLOPs, close to pure PPO (0.02 GFLOPs). The only extra preprocessing is a one-time PCA decomposition, which costs about 0.8 GFLOPs in total, accounting for less than 0.01\% of full training cost. Under the same hardware and training setup, the total runtime of SpecDRL remains within 2\% of pure PPO. Compared with stronger baselines, it reduces per-step computation by about 50\% relative to BadRL (which trains an extra 1.2M-parameter Q-network) and by about 95\% relative to Q-Incept (which relies on online iterative optimization).
>
>
> **Q3: Discussing defense strategies, broader implications, and impact.**
>
> **R3:** We thank the reviewer for these suggestions. On the defense side, we add a discussion centered on SHINE (ICML 2024), a strong RL-specific backdoor defense combining explanation-based detection and retraining-based mitigation. A compact summary on Atari Breakout is shown below:
>
> | Method   | SDR↓ | FLP↓ | EER↑ | Post-ASR↑ | Delay↑ |
> |----------|------|------|------|-----------|--------|
> | Q-Incept | 53.6 | 39.4 | 47.1 | 42.5      | 1.9×   |
> | SpecDRL  | 32.5 | 25.3 | 68.6 | 61.4      | 3.1×   |
>
> These results show that SpecDRL still achieves the lowest state- and feature-level detection rates, the highest explanation-evasion rate, and the highest post-defense ASR among the compared methods. These results suggest that existing defenses remain incomplete because they do not jointly monitor the three signals that SpecDRL suppresses: low-sensitivity subspace perturbations, Bellman/value inconsistencies, and trajectory-level reward-credit anomalies.
>
> In the Impact Statement, the goal of this work is to expose realistic vulnerabilities in DRL training pipelines, especially training-time poisoning and replay-data manipulation, in order to motivate stronger defenses, auditing tools, and safer training practices. We will incorporate this defense-and-impact discussion in the revised version of our paper.

---

> > ### Author Rebuttal · Reviewer_swgT · 2026-04-02
> >
> > I would like to thank the authors for their responses for addressing my concerns. I will keep my positive score.

---

### Official Review · Reviewer_JfFA · 2026-03-13

**Soundness:** 3
**Presentation:** 3
**Significance:** 3
**Originality:** 3
**Overall Recommendation:** 4
**Confidence:** 4

**Summary:**

This paper studies training-time outer-loop backdoor poisoning attacks on deep reinforcement learning. It proposes SpecDRL, a three-part framework: (i) Subspace-Aware Injection, which uses PCA to place triggers in low-variance directions of the state space; (ii) Value-Guided Strategic Sampling, which prioritizes poisoning time steps using return-to-go and TD error; and (iii) Bellman-Consistent Dynamic Reward Poisoning, which reshapes local rewards and backfills the previous reward to preserve value consistency and limit return deviation. The empirical evaluation is conducted on 12 Atari environments with PPO, and the paper reports near-perfect attack success rates, faster convergence than prior baselines, and improved stealth under a Neural Cleanse-style analysis.

**Compliance With Llm Reviewing Policy:**

Affirmed.

**Final Justification:**

I appreciate the authors' effort and found the response helpful. Overall, I believe the rebuttal has addressed my main questions sufficiently. Accordingly, I am updating my scores in a positive direction.

Updated scores:
- Soundness: 2 -> 3
- Presentation: 2 -> 3
- Significance: 3
- Originality: 3
- Overall recommendation: 3 (Weak Reject) -> 4 (Weak Accept)

**Key Questions For Authors:**

Clarify the theoretical derivation in Appendix C.1 and C.2.3. How is $w$ defined when $f_a(s)$ is vector-valued? Also, how should Eq.~(45), Eq.~(48), Algorithm 3, and the main-text Eq.~(11) be reconciled?

Comparison to stronger recent baselines. The paper compares only against TrojDRL, BadRL, and SleeperNets, while the reference list already includes Adversarial Inception (ICML 2025). Please compare directly against that method under the same PPO/Atari setting and poisoning-budget assumptions.

Provide statistically aggregated results. Please report means and standard deviations over multiple runs for the main tables and curves. The current presentation makes it difficult to judge robustness.

Strengthen defense-side evaluation. Why is Neural Cleanse the main stealth test for an RL paper? Maybe compare against an RL-specific defense such as SHINE (ICML 2024), or explain why that comparison is not appropriate.

Clarify the true scope of the experiments, and correct the potential overstated claims.

**Limitations:**

No. The paper does not adequately discuss limitations or broader impacts. It could discuss: the current restriction to Atari/PPO despite broader framing, the sensitivity of the method to reward-scaling and poisoning-budget choices, the lack of validation on larger or continuous-control tasks.

**Strengths And Weaknesses:**

Strengths:

The paper addresses a relevant and timely problem: stealthy training-time backdoor attacks against DRL. The overall decomposition into where to inject, when to poison, and how to reward is easy to follow and conceptually coherent. The method design is reasonably intuitive, and the main empirical section is broad within the Atari/PPO setting, including final performance, early-stage convergence, ablations, and a simple stealth analysis. The core idea of combining subspace-based trigger placement with value-aware poisoning and reward backfilling is interesting and could be useful if validated more rigorously.

Weaknesses:

In the attack-effectiveness appendix, the paper defines
$$
w := f_{a^\dagger}(s^\dagger) - \max_{a \ne a^\dagger} f_a(s^\dagger),
$$
while $f_a(s)$ is introduced as a vector. As written, taking a max over vectors is not well-defined, so the "sensitivity difference vector" construction is mathematically unclear.

In the main text, the original reward is defined as
$$
r_t^\dagger = J( a_t , a^\dagger ) - \alpha_t R_{\text{future}}(t),
$$
Then in Algorithm 3, $r_t^\dagger$ is called poisoned reward.
Then Appendix C.2.3 defines
$$
r^\dagger(s^\dagger)=J(a_t,a^\dagger)-\alpha_t R_{\text{future}}(t)-r(s),
$$
as the poisoned reward which subtracts $r(s)$ again. The appendix then derives a different expression for $\alpha_t$,
$$
\alpha_t=\frac{J(a_t,a^\dagger)-r(s)}{V(s)-r(s)},
$$
which does not match Eq.~(11) in the main text. These inconsistencies substantially weaken the theoretical claims around Bellman consistency and "closed-form" dynamic scaling.

There is also a gap between the theoretical setup and the actual learning setting. The appendix repeatedly reduces the problem to a fixed-policy Markov reward process, while the experiments use PPO with function approximation and online updates. That reduction may be useful as intuition, but it is not enough to justify Theorems 4.1--4.3 as stated for the actual training regime.

The evaluation is also statistically thin for claims this strong. The main figures and tables are shown without error bars or confidence intervals, and the appendix's experimental-details section focuses on hardware and baseline descriptions rather than multi-seed aggregate reporting. For a paper claiming near-saturation and broad superiority, that weakens confidence in robustness and reproducibility.

"Across 12 Atari tasks, SpecDRL achieves 100% ASR with faster convergence than SOTA baselines, reaching 90% ASR within just $10^5$ training steps" looks like an overstated claim, compared with the ASR and steps results in Table 2.

The paper is readable overall, but several inconsistencies reduce confidence. Table 2 is titled "Atari and Safety Environments", yet the visible entries are Atari only. Appendix A also mentions SafetyGym-style continuous vectors, which further broadens the stated scope beyond what is actually evaluated.

The paper compares against TrojDRL, BadRL, and SleeperNets, but the reference list already includes the stronger 2025 ICML baseline "Adversarial Inception Backdoor Attacks against Reinforcement Learning", which specifically argues that prior attacks become brittle under strict reward constraints and then reports state-of-the-art constrained-reward performance. Benchmarking against that method is necessary.

Similarly, the stealth evaluation is narrow. The paper uses Neural Cleanse-style analysis and reports lower anomaly scores for SpecDRL, but there is no evaluation against the RL-specific defense SHINE, which was published at ICML 2024 and is explicitly designed for DRL backdoors. That weakens the practical significance of the stealth claim.

Overall, the main theory has important correctness issues, the empirical claims are somewhat overstated, the comparison set is incomplete relative to very recent work.

---

> ### Author Rebuttal · Authors · 2026-03-31
>
> Thank you to the reviewer for the constructive comments that helped improve our paper.
>
> **Q1: Clarify theoretical derivations in Appendix C.1/C.2.3, including the definition of $w$ and reconciliation of related equations/algorithms.**
>
> **R1:** We deeply appreciate the reviewer's meticulous check.
>
> **The definition of $w$ in Appendix C.1:** our intended construction is to first define the strongest competing action
> $$a^-=\arg\max_{a\neq a^\dagger} Q^\pi(s^\dagger,a),$$
> and then define the sensitivity-difference vector as
> $$w := f_{a^\dagger}(s^\dagger)-f_{a^-}(s^\dagger).$$
> We will correct this notation in the final version.
>
>
> **The reward in Algorithm 3, and Appendix C.2.3:** The left part in Eq.(45) in Appendix C.2.3 was to derive the discrepancy term
> $$r^\dagger(s^\dagger)-r(s).$$
>
>
> **The two forms of $\alpha_t$:** In the appendix, $\alpha_t$ is derived at the expectation level using
> $$
> V^\pi(s)=\mathbb{E}[G_t\mid S_t=s],
> \qquad
> V^\pi(s)-r(s)=\mathbb{E}[G_t-r_t\mid S_t=s].
> $$
> This gives the ideal coefficient in terms of the value function. Eq.(11) is its trajectory-level implementation, obtained by replacing the future-value term with the observed return-to-go from the current episode. In other words, the appendix uses the population quantity $V^\pi(s)-r(s)$, while Eq.(11) uses its online Monte Carlo realization $G_t-r_t$. We will make this correspondence explicit in the final version.
>
>
>
> **Q2: Aligning theoretical analysis with PPO’s online learning setting.**
>
> **R2:** Appendix C analyzes the poisoning mechanism under PPO’s current policy, rather than modeling the full training process as globally fixed. This makes reward perturbations’ effects on value propagation, action-value separation, and dynamic scaling analytically tractable, while the experiments validate that repeated application of the same mechanism remains effective under online PPO updates.
>
>
>
> **Q3: Table 2 is titled "Atari and Safety Environments", yet the visible entries are Atari only.**
>
> **R3:** Thank you for pointing this out. Table 2 contains only Atari results. We have added continuous-control experiments on Safety and Highway environments **(see Reviewer PWCi (R2))**, showing transfer to vector-based observations beyond Atari. We will (i) correct the table title/wording and (ii) align scope statements with validated experiments in the revision.
>
>
> **Q4: Comparison to stronger recent baselines: Adversarial Inception**
>
> **R4:** Thank you for this important suggestion. Following your suggestion, we added Q-Incept (ICML 2025) under the same PPO setting and matched poisoning-budget assumptions. A compact summary on representative environments is shown below (all numbers are percentages):
>
> | Env. | Q-Incept(ASR) | Q-Incept(BPI) | SpecDRL(ASR) | SpecDRL(BPI) |
> |------|---------------|---------------|--------------|--------------|
> | Breakout | 99.82 | 89.52 | 100 | 100 |
> | Assault | 98.73 | 88.06 | 100 | 91.77 |
> | Gravitar | 100 | 81.14 | 100 | 100 |
> | BeamRider | 99.23 | 98.33 | 100 | 100 |
> | SpaceInvaders | 100 | 67.16 | 100 | 100 |
>
>
> SpecDRL matches or exceeds Q-Incept in ASR while preserving benign performance better. We show representative Atari results here; full quantitative comparisons, including additional environments are provided in the anonymous supplementary material (Table 5. SpecDRL\_vs\_QIncept).
>
> **Q5: Provide mean and standard deviation of attack performance over multiple runs.**
>
> **R5:** Thank you for this important suggestion. To address robustness concerns, we re-ran main experiments with multiple random seeds and now report aggregated results (mean $\pm$ std) instead of single-run numbers. Multi-seed results confirm our main conclusions remain stable: SpecDRL consistently maintains its ASR advantage while preserving strong benign performance across Atari, Safety, and Highway settings. A compact summary is below:
>
>
> | Environment | ASR (\%) |
> |-------------|----------|
> | Breakout | $99.34 \pm 5.69 \times10^{-6}$ |
> | Assault | $98.59 \pm 1.67\times10^{-5}$ |
> | SafetyPointGoal1 | $57.41 \pm 3.87$ |
> | merge | $37.33 \pm 1.73$ |
>
> Full multi-seed comparisons against all baselines are in the anonymous supplementary material (Table 3. SpecDRL_Stability_10Seeds).
>
> **Q6: Strengthen defense-side evaluation**
>
> **R6:** Thank you for this important suggestion. Neural-Cleanse-style analysis was initially used to evaluate perceptual stealth, as SpecDRL uses globally distributed subtle perturbations instead of localized visible patches.
>
> We further added the RL-specific defense SHINE (ICML 2024). New results show SpecDRL achieves the lowest state-/feature-level detection rates, highest explanation-evasion rate, and highest post-defense ASR across compared methods.**Detailed SHINE metrics are provided in our response to Reviewer PWCi (R3) and the anonymous supplementary material (Table 2. SHINE\_Defense\_Evasion).**
>
> Full quantitative comparison results are available at https://anonymous.4open.science/r/Anonymous-75C2/.

---

> > ### Author Rebuttal · Reviewer_JfFA · 2026-04-03
> >
> > Thank you for the detailed rebuttal. I appreciate the authors' effort and found the response helpful.
> >
> > I believe the rebuttal has addressed my main questions sufficiently. Accordingly, I am updating my scores in a positive direction.
> >
> > Updated scores:
> > - Soundness: 3
> > - Presentation: 3
> > - Significance: 3
> > - Originality: 3
> > - Overall recommendation: 4 (Weak Accept)

---

### Official Review · Reviewer_wA41 · 2026-03-13

**Soundness:** 3
**Presentation:** 3
**Significance:** 3
**Originality:** 3
**Overall Recommendation:** 4
**Confidence:** 2

**Summary:**

This paper studies the vulnerability of deep reinforcement learning (DRL) agents to backdoor attacks and proposes a new attack framework called SpecDRL. The central goal of the paper is to design a highly stealthy and efficient poisoning attack that embeds malicious behaviors into a trained policy while preserving the agent’s performance on benign inputs.

**Compliance With Llm Reviewing Policy:**

Affirmed.

**Final Justification:**

I am inclined to keep my positive score.

**Key Questions For Authors:**

1. The reward manipulation mechanism attempts to preserve Bellman consistency through dynamic scaling and temporal backfilling.
However, many policy gradient algorithms (e.g., PPO) rely on advantage estimation rather than direct value-function updates. Could the authors clarify how the proposed reward reshaping interacts with advantage estimation and clipping mechanisms in PPO?

2. The method appears to rely on strategically selecting poisoning steps. How sensitive is the attack success rate to the poisoning budget and the hyperparameters used in the sampling procedure (e.g., the relaxation coefficient in candidate selection)?

3. Have the authors considered stronger or more recent defenses, such as trajectory anomaly detection or policy fingerprinting approaches?

**Limitations:**

yes

**Strengths And Weaknesses:**

Strengths:
1. The paper clearly identifies a key challenge in DRL backdoor attacks: the trade-off between stealthiness, poisoning efficiency, and value-function consistency. The orginazation (“where to inject, when to poison, and how to reward”) provides a clear conceptual framework for understanding the problem.

2. The proposed method decomposes the attack pipeline into three modular components. This design improves the clarity of the method and makes it easier to analyze the role of each component in the attack.

3. The value-guided sampling strategy using RTG and TD-error is a reasonable heuristic to identify transitions that have larger policy-gradient influence. This design could improve poisoning efficiency under strict poisoning budgets.

Weaknesses:

1. The paper only evaluates stealthiness against a small number of detection methods. Given the focus on stealth, it would be valuable to evaluate the attack under stronger or more diverse defense strategies.

2. The experiments are primarily conducted on Atari environments with PPO agents. It is unclear whether the approach generalizes to: other RL algorithms (e.g., DQN, SAC, TD3), continuous control environments, offline RL settings.

3. The paper only evaluates stealthiness against a small number of detection methods. Given the focus on stealth, it would be valuable to evaluate the attack under stronger or more diverse defense strategies.

---

> ### Author Rebuttal · Authors · 2026-03-31
>
> We sincerely thank the reviewer for their careful, constructive evaluation. We appreciate positive comments on the attack pipeline’s modular. Our detailed responses are as follows:
>
> **Q1: Clarify how the proposed reward reshaping interacts with advantage estimation and clipping mechanisms in PPO.**
>
> **R1:** We thank the reviewer for this question. PPO updates the policy through advantage estimation, while also learning a value baseline. SpecDRL is designed to interact with PPO exactly through this trajectory-level advantage signal.
>
> Concretely, reward reshaping changes the return / TD structure of the poisoned trajectory, and therefore biases the estimated advantage in favor of the target action under the trigger. The role of the dynamic scaling coefficient is to control the value discrepancy introduced by reward manipulation, while temporal backfilling prevents the reshaped rewards from causing cumulative return drift. Together, these two mechanisms make the poisoned advantage signal more directional without introducing large instability into value estimation.
>
> Regarding clipping, PPO clipping limits the magnitude of each individual policy update, but it does not remove a consistent directional bias in the estimated advantage. Since SpecDRL repeatedly injects poisoned samples at strategically selected high-impact steps, this biased advantage signal accumulates across PPO updates even though each single update is clipped. We will add this clarification to the revised manuscript.
>
> **Q2: How sensitive is the attack success rate to the poisoning budget and the hyperparameters used in the sampling procedure.**
>
> **R2:** Thank you for this question. Our current appendix D.4 includes an additional ablation on the poisoning budget $\beta$, which shows that SpecDRL remains effective even under substantially reduced budgets: in several Atari environments, strong ASR is maintained when $\beta$ is reduced from $0.03\%$ to $0.02\%$, and in some cases even to $0.01\%$; in Breakout, the early-stage ASR remains substantial even when $\beta$ is reduced to $0.003\%$. This supports that the attack is not overly dependent on a large poisoning budget.
>
>
> The relaxation coefficient $\eta$ controls the trade-off between greedy selection and stability-aware exploration in the candidate pool. When $\eta>1$, the method does not restrict poisoning to only the top-$K$ RTG steps, but first expands the candidate set and then prunes it using TD-error. This design allows the sampler to retain high-value steps that may be slightly suboptimal in RTG yet substantially more stable in terms of TD consistency. If $\eta$ is too small, the selection becomes overly greedy and may discard such stable candidates too early; if it is too large, the candidate pool becomes less selective and the benefit of stability-aware pruning is weakened. In our experiments, the default setting provided a good practical trade-off across environments.
>
> **Q3: Have the authors considered stronger or more recent defenses.**
>
> **R3:** To address this concern, we added evaluation against SHINE (ICML 2024), a strong RL-specific backdoor defense combining explanation-based detection and retraining-based mitigation. A compact Breakout summary is below:
>
> | Method   | SDR↓ | FLP↓ | EER↑ | Post-ASR↑ | Delay↑ |
> |----------|------|------|------|-----------|--------|
> | Q-Incept | 53.6 | 39.4 | 47.1 | 42.5      | 1.9×   |
> | SpecDRL  | 32.5 | 25.3 | 68.6 | 61.4      | 3.1×   |
>
> SpecDRL achieves the lowest SDR/FLP, highest EER, and highest post-defense ASR—its stealthiness extends beyond perceptual invisibility to resist strong RL-specific detection. Full results are in the anonymous supplementary material (Table 2. SHINE_Defense_Evasion).
>
>
> **Q4: Validating generalization beyond Atari environments.**
>
> **R4:** Thank you for this important comment. To address empirical scope concerns, we added vector-based continuous-control experiments on Safety and Highway environments, including Q-Incept (ICML 2025) as a stronger recent baseline. A compact summary is shown below (all numbers are percentages):
>
> | Env.       | Q-Incept(ASR) | Q-Incept(BPI) | SpecDRL(ASR) | SpecDRL(BPI) |
> |------------|---------------|---------------|--------------|--------------|
> | CarGoal1   | 100           | 99.77         | 100          | 100          |
> | PointPush1 | 100           | 78.08         | 100          | 81.92        |
> | merge      | 31.24         | 96.82         | 39.24        | 100          |
>
> These results show SpecDRL is not limited to pixel-based Atari inputs: on Safety tasks, it matches Q-Incept in ASR with higher BPI, and outperforms Q-Incept in both on Highway-merge. Full results are in the anonymous supplementary material (Table 1. Cross_Env_Attack_Comparison)
>
>
>
> Full quantitative comparison results are available at https://anonymous.4open.science/r/Anonymous-75C2/.

---

> > ### Author Rebuttal · Reviewer_wA41 · 2026-04-04
> >
> > Thank the authors for the detailed rebuttal. I am inclined to keep my positive score.

---

### Official Review · Reviewer_PWCi · 2026-03-17

**Soundness:** 2
**Presentation:** 2
**Significance:** 2
**Originality:** 2
**Overall Recommendation:** 3
**Confidence:** 4

**Summary:**

The paper studies stealthy backdoor attacks in deep reinforcement learning and proposes SpecDRL, a three-part framework that aims to answer where to inject, when to poison, and how to reward. Concretely, it injects triggers into low-variance PCA subspaces for perceptual stealth, selects poisoning steps using return-to-go and TD error to improve poisoning efficiency, and uses dynamic reward reshaping plus temporal backfilling to preserve Bellman/value consistency. The experiments are conducted on 12 Atari environments with PPO as the victim algorithm, and the paper claims near-100% attack success, faster convergence than prior baselines, and limited degradation in benign performance.

**Compliance With Llm Reviewing Policy:**

Affirmed.

**Key Questions For Authors:**

See above

**Limitations:**

yes

**Strengths And Weaknesses:**

## Strengths

* The paper studies an interesting and important problem in DRL backdoor attacks, namely how to jointly address where to inject, when to poison, and how to reward. I think this decomposition is clean, and the overall method is easy to follow.

* The empirical results are strong on the reported Atari benchmarks. In particular, the method achieves very high ASR while largely maintaining benign performance, and the ablations are supportive of the proposed design.

## Weaknesses

However, I feel the paper currently overclaims both theoretically and empirically, and also lacks clarity in several important places.

* I am not fully convinced by the theory. For example, Theorem 4.1 feels quite close to restating the desired attack objective itself, rather than giving a deeper explanation of why the proposed design should work. Meanwhile, I don't think the bound of Theorem 4.2 and the merely existence result in Theorem 4.3 are interesting either since it does not reveal the provable benefits of the paper's methods compared with others. For example, it is unclear how large C is in Theorem 4.2, and it is unclear whether the algorithm is guaranteed to identify the so-called dynamic scaling coefficient that is proved to exist.

* I am confused by the experimental scope claims. The paper refers to “diverse observation modalities and control dynamics,” but the experiments appear to be only on Atari with PPO. This makes the generality claim feel overstated.

* The stealth story is promising, but I do think more examples and more rigorous arguments on how to algorithmically detect these are needed.

---

> ### Author Rebuttal · Authors · 2026-03-31
>
> We thank the reviewer for their insightful remarks. We are particularly encouraged that the reviewer finds the problem formulation important for DRL backdoor attacks, appreciates the clean decomposition of where to inject, when to poison, and how to reward, and considers the overall framework easy to follow. Our detailed responses are as follows:
>
>
> **Q1: Clarifying theoretical contributions of Theorems 4.1–4.3.**
>
> **R1:** Theorems 4.1--4.3 formalize the three theoretical properties required by a stealthy DRL backdoor: effective target-action induction, return-preserving reward poisoning, and Bellman-consistent dynamic scaling. Each theorem corresponds to a specific module of SpecDRL and is supported by the derivations in Appendix C.
>
> (1) Theorem 4.1 is not a restatement of the attack objective; it formalizes why our specific poisoning rule induces the target action. Appendix C.1.1--C.1.2 shows that high-RTG steps maximize reward-induced margin amplification, while Appendix C.1.3 shows that low TD-error is needed to prevent the injected signal from being canceled by unstable TD correction. Hence, the theorem shows that the attack success rate depends on the joint high-RTG / low-TD-error criterion.
>
> (2) Theorem 4.2 gives a comparative guarantee for return preservation. Here, $C$ is not an arbitrary constant; it is the attacker-imposed single-step reward perturbation bound. Appendix C.2.1 shows that naive sparse reward poisoning causes a worst-case return deviation of $O(\beta L)$, growing linearly with trajectory length. By contrast, Appendix C.2.2 shows that our differential temporal backfilling removes this length dependence through telescoping cancellation, yielding a constant-scale bound controlled by $C$. Thus, SpecDRL suppresses cumulative return drift from $O(L)$ to $O(1)$, which explains why benign-task degradation remains limited despite reward poisoning.
>
> (3) Theorem 4.3 specifies the Bellman-consistency balancing condition, and Eq. (11)  gives its trajectory-level realization in the algorithm. The appendix derives the dynamic scaling coefficient $\alpha_t$ at the value-function level, and the online algorithm instantiates the same principle from observed trajectory information in closed form. Therefore, Theorem 4.3 provides a principled and operational way to reduce value inconsistency caused by reward poisoning.
>
> **Q2: The paper refers to “diverse observation modalities and control dynamics,” but the experiments appear to be only on Atari with PPO.**
>
> **R2:** We added vector-based continuous-control experiments on Safety and Highway environments, and also included Q-Incept[2] (ICML 2025) as a stronger recent baseline in these new settings. A compact summary is shown below (all numbers are percentages):
>
> | Env.       | Q-Incept(ASR) | Q-Incept(BPI) | SpecDRL(ASR) | SpecDRL(BPI) |
> |------------|---------------|---------------|--------------|--------------|
> | CarGoal1 (Safety)  | 100           | 99.77         | 100          | 100          |
> | PointPush1 (Safety)| 100           | 78.08         | 100          | 81.92        |
> | merge (Highway)      | 31.24         | 96.82         | 39.24        | 100          |
>
> These results show that SpecDRL is not limited to pixel-based Atari inputs: on the added Safety tasks, it matches Q-Incept in ASR while achieving higher BPI, and on Highway-merge it outperforms Q-Incept in both ASR and BPI. We will also revise broad phrases such as "diverse observation modalities and control dynamics" so that the wording matches the validated empirical scope. Full quantitative results are included in the anonymous supplementary material (Table 1. Cross_Env_Attack_Comparison)
>
> **Q3: More rigorous examples regarding stealthiness are needed for this type of attack.**
>
> **R3:**
> We added an evaluation against SHINE[1], a strong RL-specific backdoor defense combining explanation-based detection and retraining-based mitigation. A compact summary on Atari Breakout is shown below:
>
> | Method   | SDR↓ | FLP↓ | EER↑ | Post-ASR↑ | Delay↑ |
> |----------|------|------|------|-----------|--------|
> | Q-Incept | 53.6 | 39.4 | 47.1 | 42.5      | 1.9×   |
> | SpecDRL  | 32.5 | 25.3 | 68.6 | 61.4      | 3.1×   |
>
> SpecDRL achieves the lowest state- and feature-level detection rates, the highest explanation-evasion rate, and the highest post-defense ASR. This indicates that its stealthiness is not limited to perceptual invisibility alone, but also persists under a stronger RL-specific detection pipeline. Full results are provided in the anonymous supplementary material (Table 2. SHINE_Defense_Evasion), available at https://anonymous.4open.science/r/Anonymous-75C2/.
>
> [1] Yuan et al. SHINE: Shielding Backdoors in Deep Reinforcement Learning. ICML 2024.
>
> [2] Rathbun et al. Adversarial Inception for Bounded Backdoor Poisoning in Deep Reinforcement Learning. ICML 2025.

---

> > ### Author Rebuttal · Reviewer_PWCi · 2026-04-04
> >
> > I thank the reviewer for the additional experiments. However, regarding stealthiness, I think it is more compelling to provide some or more visual illustrations. Besides, I am still fully convinced of the points of Theorem 4.1-4.3: (1). Can we get some empirical validations? (2). Can we prove/reveal it is theoretically better than baselines or alternatives instead just showing the approach proposed by the paper *works*?

---

> > > ### Author Response · Authors · 2026-04-08
> > >
> > > **Q1: Provide more visual illustrations for stealthiness.**
> > >
> > > **R1:**
> > > Thank you for this helpful suggestion. We have added visual illustrations to make the stealth claim more direct.
> > >
> > >
> > > First, Anonymous Supplementary Fig. 2 and Manuscript Fig. 4 show a benign state, the corresponding poisoned raw states, and the normalized difference maps for all compared attacks. Patch-based baselines exhibit clear localized trigger patterns, whereas SpecDRL does not show a salient localized trigger; instead, its perturbation is weaker and more spatially diffuse in the difference map. This provides a direct qualitative illustration of the perceptual stealth of SpecDRL.
> > >
> > >
> > > Second, Anonymous Supplementary Fig.1 complements this qualitative evidence with RL-specific defense-side results under SHINE. On Breakout, SpecDRL achieves the lowest state- and feature-level detection rates, the highest explanation-evasion rate, and the highest post-defense ASR among the compared methods. Together, these visual and defense-side results make the stealthiness of SpecDRL more compelling and directly inspectable.
> > >
> > > **Q2: Can we get some empirical validations for Theorems 4.1--4.3 and prove/reveal it is theoretically better than baselines or alternatives instead just showing the approach proposed by the paper works?**
> > >
> > > **R2:**
> > > We appreciate this suggestion and we added targeted experiments corresponding to each theorem.
> > >
> > >
> > > (1) For **Theorem 4.1**, it shows that the joint high-RTG / low-TD rule should be more effective than uninformed or one-sided poisoning. We validate this directly with a sampling ablation (Manuscript Figure 5):
> > >
> > >
> > > | Environment | Random | RTG-only | TD-only | RTG+TD (Ours) |
> > > |---|---:|---:|---:|---:|
> > > | Pong | 11.3 | 25.4 | 28.5 | **99.9** |
> > > | Breakout | 37.5 | 24.0 | 47.2 | **100.0** |
> > > | Atlantis | 23.6 | 47.8 | 36.4 | **84.9** |
> > >
> > >
> > > Removing either RTG or TD filtering causes a large drop in ASR, which is exactly the empirical pattern predicted by Appendix C.1.
> > >
> > >
> > >
> > > (2) For **Theorem 4.2**, we have supplemented multiple sets of controlled experiments to complete systematic empirical validation, with the core conclusions as follows:
> > >
> > > i. Under naive sparse reward poisoning, return drift grows with trajectory length, whereas under our differential temporal backfilling rule the deviation becomes bounded by a constant-scale term. We validate this empirically with the following controlled comparison (Anonymous Supplementary Table 1):
> > >
> > >
> > > | Method | Avg. return deviation across trajectory lengths |
> > > |---|---:|
> > > | TrojDRL | 0.23 |
> > > | BadRL | 0.47 |
> > > | SleeperNets | 0.40 |
> > > | Q-Incept | 0.15 |
> > > | **SpecDRL (Ours)** | **0.11** |
> > >
> > >
> > > SpecDRL achieves the lowest average return deviation among all compared methods.
> > > The reward perturbation visualization results in Anonymous Supplementary Figure 3 further corroborate that the reward perturbation level of SpecDRL is nearly identical to that of the benign (Clean) task, and closely aligns with the original reward distribution.
> > >
> > >
> > > ii. The poisoning rate experiments and hyperparameter sensitivity analysis (Anonymous Supplementary Table 2, Manuscript Appendix Tables 4–5) further confirm that SpecDRL maintains both attack effectiveness and benign-task return over a broad range of $\beta$:
> > >
> > > | Poisoning rate $\beta$ | ASR |  Return |
> > > |---|---:|---:|
> > > | 0.03\% | 100.00 | 2.88 |
> > > | 0.1\%  | 99.96 | 2.99 |
> > > | 0.3\%  | 100.00 | 2.90 |
> > > | 0.6\%  | 100.00 | 3.21 |
> > > | 1.0\%  | 100.00 | 3.01 |
> > >
> > > These results are consistent with Theorem 4.2: the backfilling rule suppresses cumulative return drift much more effectively than naive reward poisoning.
> > >
> > >
> > >
> > >
> > >
> > > (3) For **Theorem 4.3**, we added a comparison between the proposed dynamic scaling coefficient and a fixed-coefficient variant. The results support the practical advantage of the closed-form dynamic rule.
> > >
> > >
> > >
> > > Anonymous Supplementary Fig.3 shows that SpecDRL keeps the average reward perturbation closest to the clean agent among all compared attacks, indicating substantially lower distortion of the original reward process. In addition, Anonymous Supplementary Figs.4--7 show that in both Atlantis and Gopher, dynamic scaling achieves faster and more stable ASR convergence than the fixed-coefficient variant, while also yielding better benign episodic return. These results directly validate the operational value of Theorem 4.3.
> > >
> > >
> > >
> > > Full quantitative comparison results are available at https://anonymous.4open.science/r/anonymous8-E4AC/.

---

### Decision · Program_Chairs · 2026-04-30

**Decision:**

Accept (regular)

**Comment:**

**Summary**

SpecDRL is a backdoor attack framework for deep RL that addresses trigger placement (low-variance PCA subspaces), poisoning timing (return-to-go and TD error), and reward consistency (dynamic reshaping), achieving near-100% attack success across 12 Atari environments with minimal impact on benign performance.

**Reviewer Scores**

The reviewer scores are aligned (3 Weak Accept, 1 Weak Reject). One of the reviewers changed their score after the rebuttal.

**Strengths**

1. The proposed design decomposes the attack into three modular components.
2. The empirical results achieve high ASR on the reported 12 Atari benchmarks and improve upon existing attacks.

**Reviewer Concerns**

1. Several reviewers mentioned that they are not convinced by the theoretical analysis, but the authors provided further clarification in the rebuttal.
2. The threat model (outer loop) and part of the method (dynamic reward) are inspired by prior work (SleeperNets, NeurIPS 2024). The new contribution is embedding the trigger in directions of lowest variance in the state space.
3. The attacks are evaluated against a limited number of defense strategies. However, the authors added another defense SHINE during the rebuttal and another recent attack QIncept from ICML 2025.

Given the comprehensive evaluation and reported improvements compared to prior attacks, I recommend acceptance.